# Mechanically Stable β-TCP Structural Hybrid Scaffolds for Potential Bone Replacement

Matthias Ahlhelm [1,]*, Sergio H. Latorre [2], Hermann O. Mayr [2], Christiane Storch [3], Christian Freytag [4], David Werner [4], Eric Schwarzer-Fischer [4] and Michael Seidenstücker [2,]*

1    Fraunhofer Institute for Ceramic Technologies and Systems, IKTS, Winterbergstraße 28, 01277 Dresden, Germany
2    G.E.R.N. Center for Tissue Replacement, Regeneration & Neogenesis, Department of Orthopedics and Trauma Surgery, Medical Center-Albert-Ludwigs-University of Freiburg, Faculty of Medicine, Albert-Ludwigs-University of Freiburg, Hugstetter Straße 55, 79106 Freiburg, Germany; sergio.latorre@uniklinik-freiburg.de (S.H.L.); hermann.mayr@uniklinik-freiburg.de (H.O.M.)
3    Fraunhofer Institute for Cell Therapy and Immunology, IZI, Perlickstraße 1, 04103 Leipzig, Germany; christiane.storch@izi.fraunhofer.de
4    Fraunhofer Institute for Ceramic Technologies and Systems, IKTS, Maria-Reiche-Str. 2, 01109 Dresden, Germany; Christian.Freytag@ikts.fraunhofer.de (C.F.); david.werner@ikts.fraunhofer.de (D.W.); eric.schwarzer@ikts.fraunhofer.de (E.S.-F.)
*    Correspondence: matthias.ahlhelm@ikts.fraunhofer.de (M.A.); michael.seidenstuecker@uniklinik-freiburg.de (M.S.); Tel.: +49-351-888-15778 (M.A.); +49-761-270-26104 (M.S.)

**Abstract:** The authors report on the manufacturing of mechanically stable β-tricalcium phosphate (β-TCP) structural hybrid scaffolds via the combination of additive manufacturing (CerAM VPP) and Freeze Foaming for engineering a potential bone replacement. In the first step, load bearing support structures were designed via FE simulation and 3D printed by CerAM VPP. In the second step, structures were foamed-in with a porous and degradable calcium phosphate (CaP) ceramic that mimics porous *spongiosa*. For this purpose, Fraunhofer IKTS used a process known as Freeze Foaming, which allows the foaming of any powdery material and the foaming-in into near-net-shape structures. Using a joint heat treatment, both structural components fused to form a structural hybrid. This bone construct had a 25-fold increased compressive strength compared to the pure CaP Freeze Foam and excellent biocompatibility with human osteoblastic MG-63 cells when compared to a bone grafting Curasan material for benchmark.

**Keywords:** Freeze Foam; hybrid bone; biocompatibility; bone replacement

## 1. Introduction

As reported in the U.S., 7.9 million fractures occur annually, of which 5–10% develop non-unions and/or delayed unions, which are major sources of complications in the treatment of bone fractures [1]. In 2005, 17 billion dollars in medical costs were attributed to the treatment of fractures caused by osteoporosis alone. By 2025, costs are estimated to rise to 25 billion dollars [2,3]. These numbers highlight the importance of achieving early mechanical stability and load-bearing capability in long weight-bearing bones. For this reason, the successful treatment of bone defects is of great importance. Vascularity and mechanical stability need to be taken into account. The "gold standard" of many surgical techniques used to reconstruct bone for critical-sized bone defects is the use of autologous bone tissue [4]. However, the use of autografts has limitations, like donor-side morbidity, additional operations, or limited availability of tissue, as well as geometric mismatch between the harvested bone and the defect site, which can result in voids and poor integration [5,6]. Further alternative substitutes are allografts and xenografts [7]. Xenograft (animal-derived material) approaches often carry risks, like inflammation and rejection of

the transplant due to physiological incompatibility of animal organs in human beings [8]. Therefore, research activity concerning bone-grafting approaches has shifted from natural grafts to synthetic bone graft substitutes and the use of biological factors [9]. Among these materials, next to metals (e.g., titanium, titanium alloys [10]) and bioglasses [11], ceramics like calcium phosphates (CaP; e.g., tricalcium phosphates (TCP), hydroxyapatite (HAp)) as well as added active growth factor recombinant human bone morphological proteins (e.g., rhBMPs) are the typical materials of choice, either alone or in combination [9,12]. Calcium phosphate ceramics are among the most commonly used and effective synthetic bone replacement materials. For example, β-tricalcium phosphate (β-TCP) is osteoconductive and is integrated into the bone without a disturbing connective tissue layer [13,14]. This property combined with its cell-mediated resorption enables the complete regeneration of bone defects. Pores and especially micropores (0.1 to 10 μm) promote bone ingrowth and can give β-TCP osteoinductive properties [13,15–18]. From this, it can be deduced that the bone implants should have similar properties, in terms of porosity, strength and stiffness, to the piece of bone to be replaced. If the strength of the implant is too low, there is a risk that the component will fail after implantation. However, if the strength and stiffness of the implant is too high, the surrounding bone will degrade. This process is known as "stress-shielding" and, like component failure, should be avoided at all costs [19,20]. The compressive strength of spongy bone is between 2–20 MPa, depending on the literature reference. With regard to porosity, cancellous bone has values between 50–90%, which explains the low mechanical load capacity. At the same time, however, the bone becomes light and the pores enable the supply of nutrients to the bone and the removal of metabolic products [21,22]. The majority of studies on CaP scaffolds focus on bone growth in the macropores (>100 μm), where bone structures such as osteones and trabeculae can form. However, more and more studies show that micropores (<50 μm) also play an important role. Not only do they improve bone growth in the macropores, but they also provide additional space for bone growth [23]. Bone growth in the micropores offers great mechanical advantages in CaP scaffolds, as it optimizes the properties of otherwise brittle materials by further stabilizing the implant and improving load transfer.

To date, specific material combinations have been examined in order to combine the tissue engineering advantages of organic materials with the mechanical load resistance of inorganic materials. Examples of such composite biomaterials are given in [24] and [25]. The latter reports on inorganic-organic hybrid scaffolds. Polyethylene glycol (PEG) and star poly(dimethylsiloxane) were mixed with bone-like matrices collagen type I, CaP, and osteocalcin, indicating that developed hybrid gels may prove promising for osteochondral regeneration. However, the compressive strength was limited by these polymers. In addition, no specific 3D construct was achievable. Ref. [24] reports on the fabrication of porous SF/β-TCP hybrid scaffolds for bone tissue reconstruction by a freeze-drying process. The manufactured scaffolds demonstrated high porosity (>60%) with good interpore connectivity and showed good biocompatibility. However, compressive strength and modulus were relatively low (<1 MPa), and no complex 3D scaffold was achieved.

Before the development of additive manufacturing technologies, ceramic bone replacement structures were usually manufactured using so-called dip coating processes in order to be able to approximately reproduce the filigree and highly porous structures [26]. Foams made of polyurethane, for example, serve as a lost form in this process. The polymer foam is cut to size and dipped into a ceramic slurry, which penetrates the pores. In a subsequent processing step, the foam mold is then burned out, and the scaffold is sintered. Although this approach can be used to produce highly porous structures, the resulting geometry is not greatly influenced but is rather predetermined by the (PU)foam. This disadvantage is overcome by the use of specific direct foaming or additive manufacturing technologies, since the mold geometry can be specifically modeled or is not needed at all. An example of an additively manufactured SiO$_2$- and zinc-doped β-TCP scaffold is given in [27]. Although the results indicate that addition of dopants to the TCP scaffolds enhanced early stages of bone formation and implant fixation when compared to pure

TCP alone in a rabbit tibia model, the compressive strength of the achieved scaffolds only amounted to around 6 MPa.

The human bone consists of a dense and solid outer shell (*Substantia corticalis*) and an inner porous filling (*Substantia spongiosa*). In order to be able to reproduce such bone architectures with different structures, which could be used as implants in the future, two technologies were recently intelligently combined. The outer shell of the bone was produced using a commercial three-dimensional (3D) printer, and the sponge-like inner bone structure was reproduced by a ceramic foam [28,29]. For the foam production, so-called Freeze Foaming was used. In this approach, in a freeze dryer, the ambient pressure around an aqueous ceramic suspension is lowered, causing the suspension to first foam and then to suddenly freeze. Ongoing pressure reduction lets the frozen water sublimate, i.e., it evaporates without becoming liquid beforehand. A subsequent heat treatment produces a solid ceramic foam. In the next step, the porous bone-like structures are fitted to a customized, complex outer ceramic shell and, thereby, made mechanically more stable. This is where additive manufacturing (AM) comes into play. One of the best-known processes in AM is the conventional stereolithography (SLA) process. This process basically allows photopolymerizable suspensions, which are filled with ceramic particles, to be cured by a UV laser. Today, the commercially available material portfolio using lithography-based ceramic manufacturing (LCM) for high-performance components also works with β-TCP, thus playing a role in this contribution. The LCM technology as a projection-based (PSL) top-down process with a light source in the blue range (452–465 nm) is representative of the so-called ceramic additive manufacturing vat photopolymerization (CerAM VPP) process (Ceramic Additive Manufacturing Vat Photopolymerization). This allows a digital micro-mirror unit, which splits a light beam into individual pixels and then projects a digital image pixel-by-pixel onto the building platform. This makes it possible to image the entire contour of the component cross-section without a mask. Thus, layer by layer, a complex 3D structure is created. In a last hybridization step, the two methods can be combined to produce porous-dense, graded, structural hybrids by a joint sintering process. However, it is possible to not only foam within additively manufactured structures but also to foam them in. This solution makes it possible to provide a porous and sponge-like scaffold as the lead structure for cells to grow into, and at the same time, AM parts serve as load-bearing support structures. In this current study, advanced scaffolds made of β-TCP were manufactured and analyzed in terms of their biocompatibility in vitro and in vivo and tested for their mechanical behavior. The authors postulated that such a complex inorganic hybrid structure, due to the combination of load-bearing support and porous cell-ingrowth-allowing interior, will eventually allow the manufacturing of bone-like mechanically stable implants that are potentially applicable for long-bone defects.

## 2. Materials and Methods

### 2.1. Freeze Foaming

Hydroxyapatite (Sigma-Aldrich, now Merck KGaA, Darmstadt, Germany; BET = 70.01 $m^2/g$, $d_{50}$ = 2.64 µm) was chosen as the raw material. Prior to suspension, it was calcined at 900 °C for 2 h to reduce the BET (now only 5.9 $m^2/g$). The ceramic suspensions consisted of water, Dolapix CE 64 (Co. Zschimmer & Schwarz Mohsdorf GmbH & Co. KG, Burgstädt, Germany) as a dispersing agent, the ceramic powder, polyvinyl alcohol as the binder and a rheological modifier (Tafigel AP15, Co. Münzing Chemie GmbH, Heilbronn, Germany) in combination with 2-Amino-2-methyl-1-propanol—AMP (Merck KGaA, 64,271 Darmstadt, Germany) for pH adjustment. The following processing route was used: 49 wt.% deionized water, 1.3 wt.% polyvinyl alcoholic binder, hydroxyapatite and 4.6 wt.% dispersing agent, referring to powder content, were mixed in a centrifugal vacuum mixer (ARV310, Thinky Corporation, Fukuoka, Japan). To disperse the particles and reduce agglomeration, the mixture was exposed to a high stirring rate (2000 rpm, mixing time 1 min, with 3 $ZrO_2$ mixing spheres of 10 mm diameter). The spheres were then separated, and 1.9 wt.% rheological modifier together with 1.5% wt. AMP was added. To

distribute the modifier, the suspension was mixed for 2 min at 1500 rpm. Afterwards, the suspensions were filled into specific molds (see Section 2.3) and transferred to a freeze dryer (Lyo Alpha 2–4, LSCplus, Co. Martin Christ Gefriertrocknungsanlagen GmbH, Osterode, Germany) for Freeze Foaming.

Freeze Foaming for In Vivo Studies

For in vivo studies, the potential of using the Freeze Foaming process and manufacturing structural hybrids was assessed for the manufacture of artificial rat bones. A femoral bone from an 11-week-old rat was scanned using computer tomography followed by CAD file reverse-engineering. From this, a femoral bone segment was chosen, and single Freeze Foams as well as structural hybrids were manufactured (see Section 2.4). In addition to using HAp as the initial material, $ZrO_2$ (TZ-3YS-E; Co. TOSOH with a $d_{50}$ = 0.7 µm) was used as a bioinert counterpart for manufacturing porous Freeze Foams. In doing so, we hoped to be able to differentiate between material effects and structural effects regarding possible in vitro/vivo results. Ideally, the Freeze Foam's characteristic porous structure/pore morphology would perform independently from the bioceramic materials.

### 2.2. FE Analysis and Material Failure Model

For providing a mechanical stability of porous bioceramics sufficient to address long-bone defects, a support structure is needed. This is offered either by providing a shell-like structure that mimics a complete artificial *corticalis* and/or by an outside, broad, accessible support structure that provides adequate strength. Both approaches are presented. However, the focus was on an outside accessible support. A simple column geometry was chosen as such support (Figure 1a, left). Its additive manufacturing by the VPP process should be uncomplicated and provide required mechanical support as well as load balance for inside lying foams. A FE (Finite Element) analysis was made (ANSYS v. 2020 R2, ANSYS Inc., Pittsburgh, PA, USA) to approximate mechanical loads appearing in the structure and, additionally, to identify possible locations of material failure. Therefore, a geometrical model was created with respect to the existing symmetry conditions (Figure 1a, right).

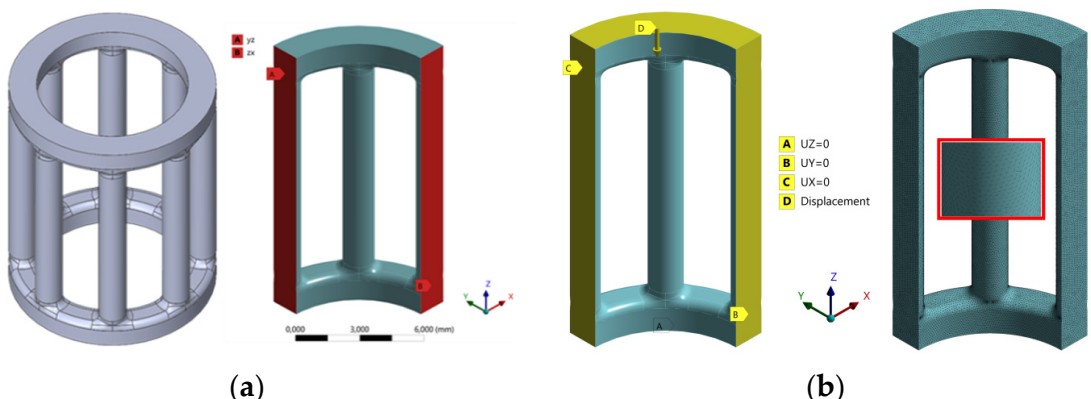

**(a)**  **(b)**

**Figure 1.** Simple column structure and boundary conditions: (**a**) column structure (left) for simulation using geometry regarding symmetry planes (right); (**b**) boundary conditions (left); meshed geometry (right).

Corresponding boundary conditions were applied to the model (Figure 1b, left), which implies fixation of translational DOF (Degree of Freedom) in the z-direction at the bottom face, symmetry conditions in the x- and y-direction and a given displacement at the top face of the structure. A displacement instead of a pressure load was used to match experimental conditions where a press specifies a defined displacement, and the resulting pressure (reaction force) was measured. Displacement values for the model were selected to achieve reaction forces of about 800 N for 1/4 of the geometry and 3200 N for the complete geometry. This load was well estimated because experimental

data indeed showed that the full column structure withstood average loads up to 3200 N (see Section 3.2 on mechanical characterization). Table 1 shows the displacements and reaction forces used for the simulation. Three load cases were chosen to simulate the complex deformation of the ceramic VPP structure. Because of the well-known brittleness of ceramic hydroxyapatite/TCP [30], the load was slowly increased, and the deformation was determined starting from a 780 N reaction force. Material failure should occur suddenly at a certain point, as shown by simulation results.

**Table 1.** The displacement, reaction forces and respective maximum principal stresses used in the structure.

| Load Case | Displacement/mm | Reaction Force/N | Maximum Principal Stress/MPa |
|---|---|---|---|
| 1st | −0.41 | 780 | 61 |
| 2nd | −0.45 | 806 | 80 |
| 3rd | −0.48 | 809 | 128 |

The chosen material for the model geometry was $\beta$-TCP. Young's modulus can be assumed to be 5.37 GPa at a porosity of 18% (previously measured [31]). Previously measured porosities of sintered VPP column structures varied between 5 and 20%, because process conditions, not yet thoroughly optimized to the $\beta$-TCP suspension, led to micro delamination between layers and/or cavities within the bulk material. For simulations, a porosity of 5% was selected, representing a stable and load-bearing material. Young's modulus, depending on porosity, can be estimated using the following Equation [32]:

$$E(\phi) = E_0(1 - 2\phi) \tag{1}$$

where $E(0.05)$ = 7.552 GPa for porosity of 5% ($E$ is Young's modulus and $\phi$ is porosity). There was no given data for Poisson's ratio. To be reasonable, 0.22 was chosen. The model geometry was discretized with a tetrahedral mesh of an appropriate density to get reliable results for a mechanical solution (Figure 1b, right). It was solved considering non-linear mechanical behavior (large deflection effects) and a high-resolution load stepping for simulation runtime to capture complex deformation of the model. Whether those material failure studies fit the experimental compressive strength tests is further shown in Section 3.2.

### 2.3. CerAM VPP

As bioceramic material, the same calcined HAp was used as for the Freeze Foams. HAp (content: 40 vol.-%) was dispersed in a fluid (polyethylene glycol, Sigma-Aldrich, now Merck KGaA, Darmstadt, Germany) with a dispersing agent (BYK-Chemie), and various monomers were used a as binder (a mixture of acrylic resins) and a photoiniator (combination of a camphor derivate and an amine). A planetary centrifugal high-speed vacuum mixer (Thinky ARV310, Thinky Corporation, Tokyo, Japan) was used for the stepwise (three times 5 min at 2000 rpm) preparation of the suspensions. Following CerAM VPP manufacturing, the column geometry as defined in Section 2.2 was chosen.

### 2.4. Mold Filling, Hybridization and Part Characterization

For achieving the hybrid parts, cylindrical rubber molds were made, in which the CerAM VPP parts fit in very closely. In the first step, the ceramic suspension filled the molds. Then the column structure was pushed into the cavity. All molds were then transferred to the freeze dryer and foamed at once. A crucial step in creating the structural hybrids is the adjustment of the shrinkage of the two different structures. At the beginning, different suspensions were developed having different contents of water, rheological modifier and binder. With those suspensions, Freeze Foams were manufactured, which shrank between 30–46% (determined by thermo dilatometry DIL 402 C/7/G Netzsch-Gerätebau, Selb, Wunsiedel, Germany). The VPP-manufactured column structure shrank

around 30% altogether. However, it was found that the VPP part shrank around 5.4% at the beginning of the heating process (the debinding), whereas the foam did not. As a result, the foams would have shrunk onto the VPP part, leading to a part failure. To compensate for the overall shrinkage, the VPP columns were pre-sintered prior to being used for in situ Freeze Foaming. This pre-sintering amounted to a shrinkage of around 5% volume. Thus, carefully adjusted for shrinkage, the hybrid parts were sintered at 1250 K (+50 K overheating effect) for 1.5 h. Afterwards, the column-including foams were dismantled and evaluated regarding porosity and microstructure. It must be noted that, after sintering, the initial hydroxyapatite was changed to β-TCP. Among many other references reporting about the transition of HA to TCP during heat treatment, similar Freeze Foams with the same initial HA powder were analyzed via XRD in a previous work [22], showing the HAp to TCP transformation. For microstructure analysis, the resulting Freeze Foams were characterized by SEM (Ultra 55, Co. Carl Zeiss, Oberkochen, Germany). By measuring the height and diameter of three different foam positions of manufactured Freeze Foams and deriving the average, geometrical porosities were calculated according to ($P$ = porosity, $\rho_{th}$ = theoretical density, $\rho_{bulk}$ = bulk density):

$$P = 1 - (\rho_{th}/\rho_{bulk}) \tag{2}$$

In addition, porosity was determined via a foam structure analysis tool based on computer tomographic images of the manufactured parts. The allocation of that 3D volumetric pore morphology information (foam cell size) was managed using VGStudio Max v3.0 (Volume Graphics GmbH, Heidelberg, Germany). For X-ray computed tomography, a CT-Compact (Procon X-ray, max. 150 kV power) was used. The universal electromechanical testing machine Instron 8562 (Norwood, MA, USA) was used for the compression strength tests (load cell 10 kN, 1-Taster). To determine the surface roughness, the hybrid foam samples and the Cerasorb M (Kleinostheim, Germany) control were examined using a KEYENCE 3D Laser Scanning Microscope VK-X210 (Keyence, Osaka, Japan). The surface roughness (Sa) was determined using KEYENCE VK analysis software version 3.5.0.0. At least 3 different samples of each were analyzed. Five different positions were determined for each sample. The specimens were measured using 400x magnification.

### 2.5. In Vitro Biocompatibility

For comparison purposes, commercially available β-TCP ceramics, cylindrical Cerasorb M moldings with a diameter of 7 mm and a length of 25 mm, were purchased from Curasan (Kleinostheim, Germany). In all experiments, we used our hybrid foam scaffolds, consisting of (TCP) and, for comparison, β-TCP scaffold (Cerasorb M) cubes with an edge length of 15 mm and regular parallel macropores (1.2–1.4 mm), purchased from Curasan (Kleinostheim, Germany). Biocompatibility experiments were performed using the human osteoblastic MG-63 cells (ATCC CRL 1427). The cells were first thawed from the liquid nitrogen tank (at −196 °C) in passage 15 and cultured in a Dulbecco's Modified Eagle Medium (DMEM) with an F12 nutrient content and additives consisting of 1% penicillin/streptomycin (P/S) and 10% fetal bovine serum (FBS). Cells were maintained in a New Brunswick Galaxy 170R incubator (Eppendorf, Hamburg, Germany) at 37 °C, with a $CO_2$ saturation of 5%. The cells were passaged twice a week and then split 1:10 and 1:5. For all biocompatibility tests and experiments with SBF, the scaffolds were heat-sterilized at 200 °C for 4 h in a UF500 drying oven (Memmert, Schwabach, Germany). Each experiment was repeated 3 times.

#### 2.5.1. Live/Dead Assay

The live/dead examinations were performed after 3, 7, and 10 days. Three samples per scaffold (DD, Curasan) per time period were placed in cell culture plates. Subsequently, 50,000 cells each, which were in 200 µL of medium, were placed directly onto the samples and incubated for 2 h at 37 °C, with a $CO_2$ saturation of 5% in the incubator so that the cells could adhere to the surface of the samples. After two hours, 2.5 mL of a DMEM-F12 (Art.

No. BE12-719F, Lonza, Basel, Switzerland) complete medium was added to each well and incubated in the incubator for a defined time (3, 7, and 10 days). After this, the samples were prepared for staining. The staining solution was first prepared by adding 2 mL of DPBS (Art. No. 14190-094, Gibco, Grand Island, NE, USA) to a Falcon and 4 µL of Ethidium Homodimer III (Eth D-III) solution. The solution was then mixed. Then, 1 µL of calcein dye was added and mixed again. Finally, the prepared solution was covered with aluminum foil due to the sensitive fluorescent dye. For staining after the first cultivation, the medium was removed, and the cells were washed to eliminate serum esterase activity. Subsequently, the cells were stained according to the protocol [19]. After incubation, the cells were inspected under a fluorescence microscope. For evaluation, images were taken with an Olympus fluorescence microscope (BX51, Olympus, Tokyo, Japan) from five different positions, with $5\times$ and $10\times$ magnification on the scaffolds. Then, the ceramics were cut horizontally and viewed at the same three positions with the known magnifications. Living cells fluoresced green under blue light, and dead cells fluoresced red.

### 2.5.2. Cell Proliferation Assay

Three samples of each of the differently sized scaffolds were examined after 3, 7 and 10 days using the WST-1 test. A Nunc™ Thermanox™ Coverslip (Thermo Fisher Scientific, Waltham, MA, USA) membrane served as the positive control. All samples and controls were equally covered with 50,000 cells in 200 µL. The cells were incubated for 2 h at 37 °C, with a $CO_2$ saturation of 5% in the incubator so that they could adhere to the surface of the sample. At the end of this period, 2.5 mL of the DMEM-F12 complete medium was added to each sample and incubated. A medium change with the DMEM-F12 with the 10% FBS and 1% P/S additives was performed for days 7 and 10. The plate from day 3 was prepared for the WST evaluation. The medium was aspirated, and the wells were washed three times with PBS. The samples and the Thermanox coverslips were then transferred to a new well, and then 2.5 mL of the DMEM-F12 phenol red free (Art. No. 11039-021, Gibco, Grand Island, NE, USA) with the 1% P/S and 1% FBS additives were added to the wells with the sample (TCP + R). A total of 400 microliters of the medium was added to the previously used empty sample wells (TCP), positive control (C + R), empty control well (C+) and the blank. The blank contained only the DMEM medium without phenol red and was measured to account for background absorption. A 10% WST reagent (Art. No. 05015944001, Roche, Basel, Switzerland) was added to the corresponding volume of medium. Thus, 250 µL WST was added to the wells with sample (TCP + R), and 40 µL was added to the old wells (TCP and C+), the blank wells and the positive control (C+). This was incubated in an incubator at 37 °C for 2 h. After this time, the liquids were transferred into a 96-well plate. Three times in a row, 100 µL of each solution was added to the wells. The absorption was then measured at 450 nm using a Spectrostar Nano microplate reader (BMG Labtech, Ortenberg, Germany). The experiment was performed at least three times for each time point (3, 7, and 10 days).

### 2.5.3. Lactate Dehydrogenase (LDH) Assay

The scaffolds for use in the lactate dehydrogenase LDH experiment were seeded in three 12-well plates. Each experiment assessed three scaffolds from each size, three Thermanox coverslips each as controls, a positive control, a negative control, and a blank to account for background absorbance in the ELISA reader. The experiments were repeated at least three times. A 200 µL cell solution containing 50,000 cells was seeded onto each scaffold, and a 100 µL cell solution containing 50,000 cells was seeded onto the Thermanox coverslips and additionally into two empty wells to act as the positive and negative controls, respectively. One well was left empty for use as a blank. The well plate was placed in an incubator at 37 °C with 5% $CO_2$ for 2 h. Following incubation, 2.5 mL of DMEM-F12 phenol red free with the 1% P/S and 1% FBS additives was added into the samples wells and negative control wells. Since FBS itself contains LDH, a concentration of 10% in the medium might have triggered background absorption. Therefore, only a concentration

of 1% FBS was added to the medium. For the positive controls, 1% Triton X 100 (Art. No. X100, Sigma Aldrich, Saint Louis, MO, USA) was added to the DMEM-F12 medium with 1% P/S and 1% FBS to 100% to kill the cells. The LDH experiments were carried out at 24, 48 and 72 h following seeding, and the same procedure was repeated at each interval. Three 100 μL samples were taken from each well into a 96-well plate. An LDH reagent (100 μL) was added to each well in use, and the plate was incubated in darkness at room temperature for 30 min. Following incubation, the plate was placed in a Spectrostar Nano microplate reader, and absorbance was measured at a λ of 490 nm with a reference λ of 600 nm.

### 2.5.4. GIEMSA Staining

MG-63 cells were seeded onto the samples analogous to Sections 2.5.1–2.5.3 and stained after 3, 7 and 10 days with GIEMSA solution (GIEMSA Azure Eosin Methylene Blue, Merck). For this purpose, the samples were washed with PBS and incubated with 1 mL GIEMSA solution (diluted 1:10 with deionized water) for 10 min at room temperature. Samples were subsequently rinsed with deionized water. Microscopy was performed on an OLYMPUS SZ-61 stereo microscope.

### 2.6. In Vivo Preparations

A total of 30 female Wistar rats aged between 12 and 15 weeks were used to test the TCP implants. They were divided in six groups of 5 rats each. One group was a negative control (SHAM-surgery without any implant), and the zirconium oxide group served as a positive control along with the four experimental groups. Before we started, the animals were housed in the IVC cages, with two daily feedings and water *ad libitum*, in the Animal Testing Center of the Fraunhofer IZI. Animals were examined for release 5 days before surgery. On surgery day, the rats were anesthetized with a fully antagonizable cocktail containing Medetomidin (0.15 mg/kg), Midazolam (2.0 mg/kg) and Fentanyl (0.005 mg/kg) i.m. The artificial bones were implanted into a prepared subdermal pocket of the rat's flanks and closed with clips. The sham was treated the same way but was put into the subderm of the flank. The antagonizing was done with a cocktail containing Atipamezol (0.75 mg/kg), Flumazenil (0.2 mg/kg) and Naloxon (0.12 mg/kg) i.m. On the surgery day, and up 2 days after this intervention, the rats were treated with meloxicam 0.2–0.5 mg/kg s.c. During the controls on day 2, 7 and 14, the rats were anesthetized in a box with 2.0–3.0% isoflurane (0.8–1.5 L/min oxygen) and kept in this unconscious status with 2.0% isoflurane (0.4–0.8 L/min oxygen). On surgery day (day 0) and every control day, 500 μL of blood was taken to test the liver (ALT, AST, GGT) and kidney values (Urea, Creatinine). The test was performed by the Clinic for Ungulates of the University of Leipzig after centrifugation of the blood at $10,000 \times g$ for 5 min at room temperature. The serum was then stored at $-20\ ^\circ$C. On days 7 and 14, we also took a fine needle biopsy of the implant location averted from the incision and fixed it with 4% PFA. On day 14, the animal tests were finalized using deep isoflurane anesthesia. In the following necropsy, tissue was removed for further investigation from the liver, kidney, spleen implant location and local lymph nodes of the implant location, which was preserved in 4% PFA. Implants were transferred into a 15 mL BlueCup with saline, photographed with a Leica camera 2.0 after 1 to 3 h and evaluated semi-objectively according to vascularization/tissue ingrowth, removability of the tissue and loss of substance/tendency for break using a numerical system. In the case of an indifferent vascularization/tissue ingrowth, we evaluated two halves of each of the bones and took the average value.

0 = no
1 = minimal
2 = minimal–moderate
3 = moderate
4 = moderate–high
5 = high

This semi-objective data and the serum parameters were collected for each group in an Excel table and evaluated via box-plot. To assess the anticipated differences of tissue attachment to the porous foam section and the denser, rather smooth CerAM VPP-manufactured surface, whole artificial rat bones were cut in half. Each half was further analyzed.

### 2.7. Statistics

The collected data were analyzed descriptively using SPSS statistics software (Version 25, IBM, Armonk, NY, USA). Based on the raw data, the mean value and the standard deviation were calculated. The Mann-Whitney U-Test was used to evaluate the differences between experimental and control samples. $p$-values < 0.05 were considered to indicate statistical significance.

### 3. Results

The subsequent figure shows one exemplary manufactured near-net shaped hybrid foam (Figure 2a, right side) consisting of the foamed-in additively manufactured support structures (Figure 2a, left side) and the porous Freeze Foam, which together make up the support structure case. Figure 2b illustrates the Curasan control. Figure 3 displays the workflow from the rat bone to the reverse-engineered CAD file to the manufactured single Freeze Foams and bioceramic artificial *corticalis* (i.e., *corticalis* case).

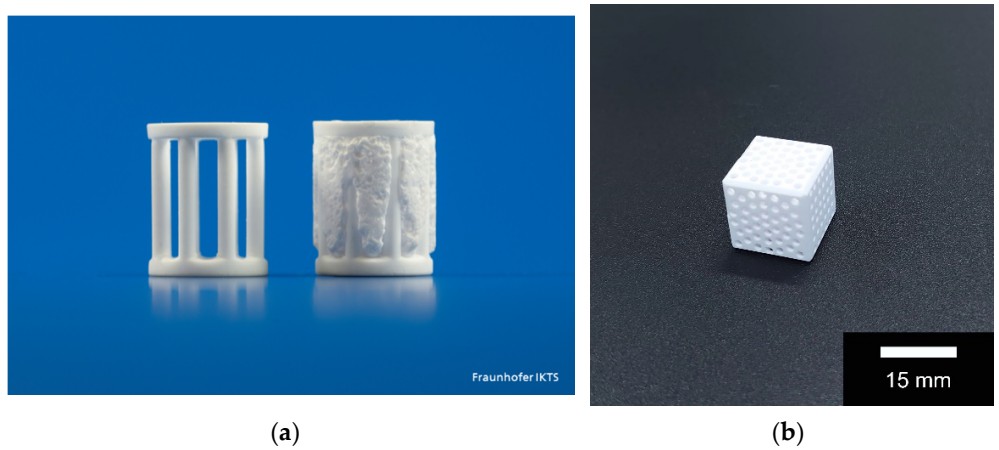

(**a**)  (**b**)

**Figure 2.** Overview samples vs. Curasan control: (**a**) VPP-manufactured column support structure (left) and hybrid with Freeze Foam enclosing the VPP support (same figure, right); (**b**) Curasan control.

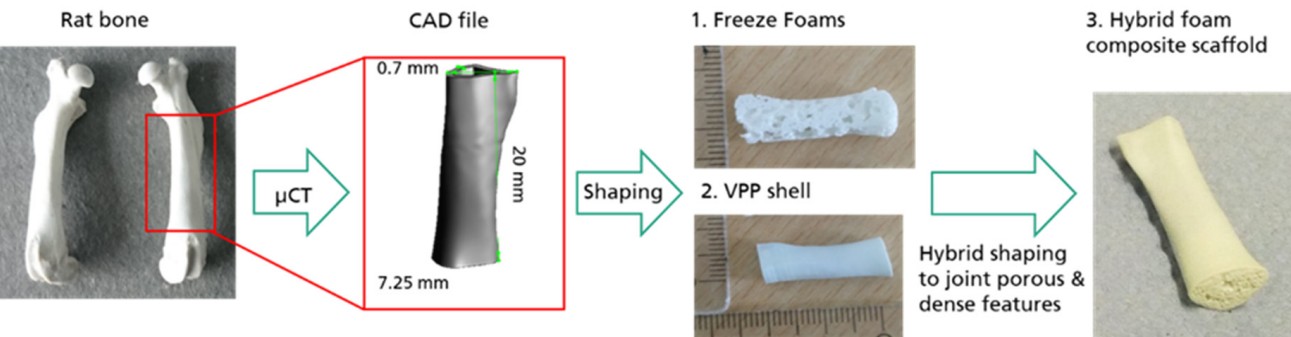

**Figure 3.** Reverse-engineering workflow from rat bone to a CAD file to the manufactured Freeze Foams and hybrid foams (artificial *corticalis* case).

### 3.1. Microstructural Characterization

Computer tomographic images of an exemplary hybrid structure confirmed that the form and material fit between the columns and the foam (Figure 4). From left to right, first the VPP column and then the hybrid, both in the green state, are displayed, followed by the sintered hybrid.

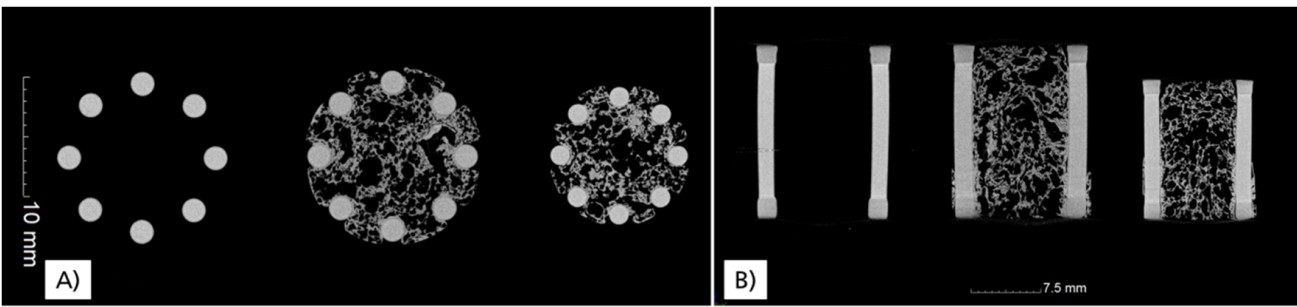

**Figure 4.** CT images of an exemplary hybrid structure; from left to right: left: VPP support structure (green state); middle: hybrid (green state); right: sintered hybrid (**A**: plan view, **B**: side view).

SEM images clearly showed the denser VPP-manufactured round column and the porous Freeze Foam (Figure 5). At the junction between them, several gaps appear. Foam and the VPP part fused together, but only partially and only in a few spots. However, this SEM analysis only shows one specific location within one hybrid structure. More hybrids need to be manufactured and examined to come to a general conclusion.

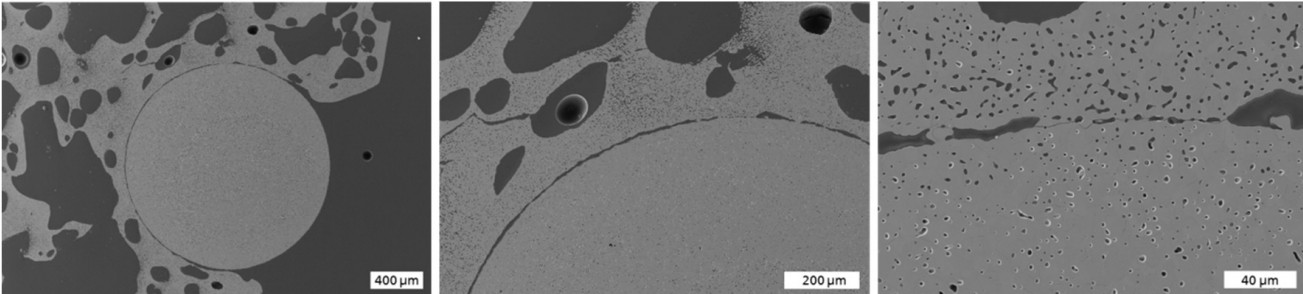

**Figure 5.** FESEM cross-cutting images displaying interface between VPP-manufactured column and Freeze Foam; magnification/HFW 35×/3334 μm (**left**); 100×/1160 μm (**middle**); 500×/242 μm (**right**); 8 kV acceleration voltage.

The gap between column and foam was measured at one location (Figure 6), which varied between 1 and 13 μm. In general, the column was much denser than the foam, with macropores of around 100 to <600 μm. A closer look at a higher magnification showed mesopores of around 1–2 μm in the foam and in the struts (Figures 5 and 6).

One hybrid foam was analyzed in the fractured view (Figure 7). The gap at the interface between the foam and column is obvious as well as the interconnected pores in the Freeze Foam. At this location, a higher magnification showed that the material fit between the foam and the column and formed the TCP microstructure, with mesopores of around 1–2 μm.

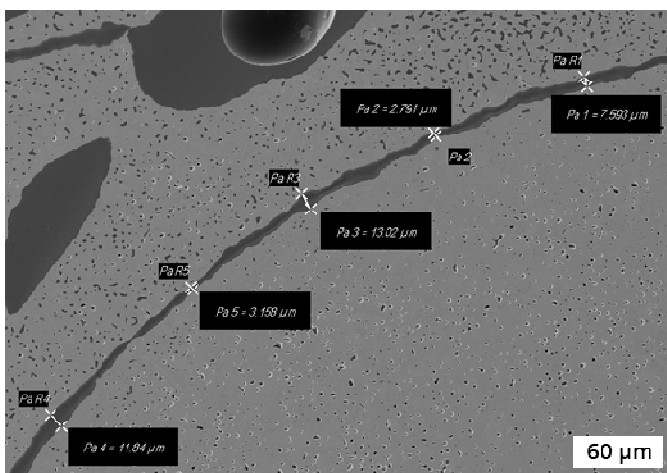

**Figure 6.** FESEM image: measured distance in the gap interface of VPP column and Freeze Foam.

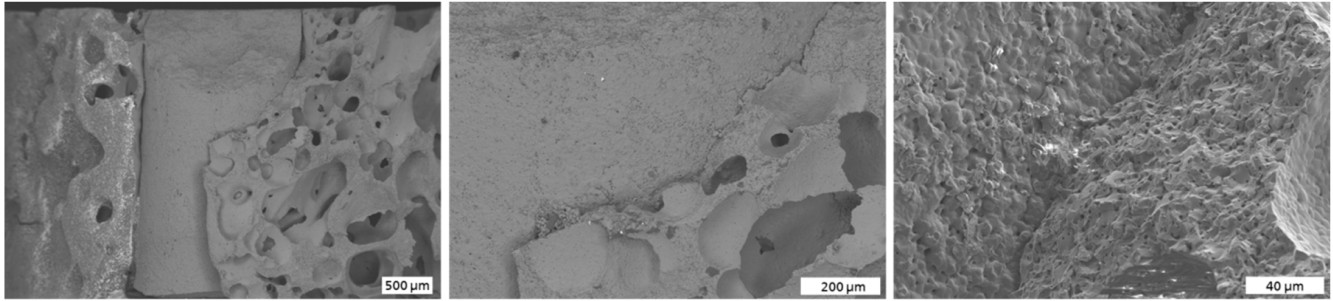

**Figure 7.** SEM fractured view of gap interface of VPP column and Freeze Foam: magnification/HFW 26×/4446 μm (**left**); 100×/1109 μm (**middle**); 500×/227 μm (**right**); 6 kV acceleration voltage.

Fractured and cross-sectioned images (Figure 8) once again show a good material fit between the column and the foam. As stated before, the interface connection was not thoroughly complete, and its state/appearance depended on the location in the hybrid (referring to each column and each column length).

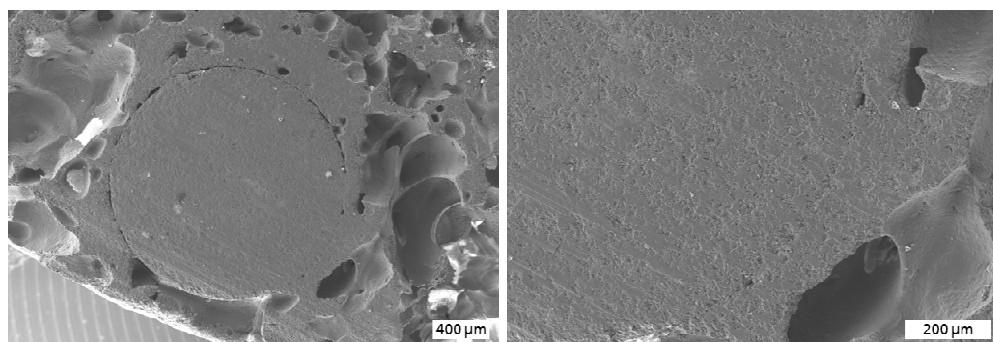

**Figure 8.** SEM fractured and cross-section view of gap interface of VPP column and Freeze Foam: magnification/HFW 35×/3550 μm (**left**); 100×/1109 μm (**right**); 8 kV acceleration voltage.

The surface roughness Sa was determined to be $5.99 \pm 1.43$ μm for the Curasan control, $3.73 \pm 1.94$ μm for the outer ceramics (column's) ring of the hybrid foam and $6.54 \pm 2.93$ μm for the inner foam structure of the hybrid foams (Figure 9).

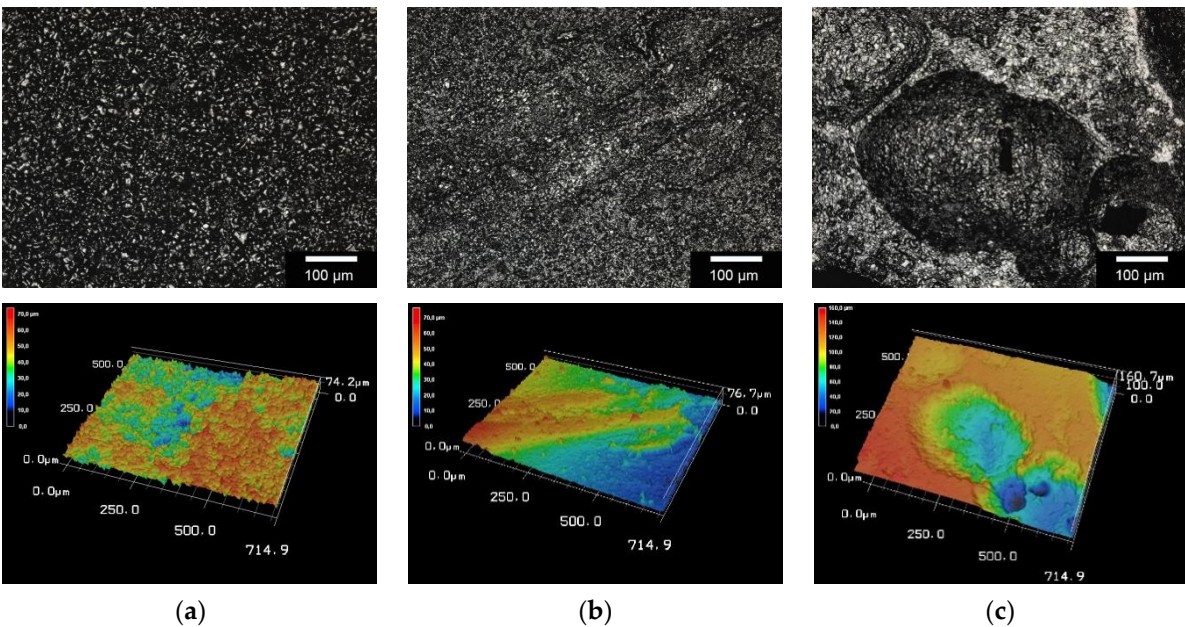

(**a**)  (**b**)  (**c**)

**Figure 9.** 3D Laser scanning image (top) and 3D reconstruction (bottom) of: (**a**) Curasan control; (**b**) Hybrid Foam, outer ceramic ring; (**c**) Hybrid Foam, inner foam. Images were taken with the KEYENCE VK-X210 3D Laser scanning microscope, 400× magnification.

### 3.2. Mechanical Characterization and Comparison to FE Simulation

One of the aims of this study was to enhance the mechanical stability of the TCP. Therefore, Freeze Foams and hybrid foams were tested for compressive strength. The following table summarizes the compressive strength (relating to the cylindrical cross-section) and porosity (geometrical and foam cells determined via foam structure analysis based on CT images) of tested samples (five each, mean values shown in Table 2).

**Table 2.** Compressive strength of manufactured components (foams, hybrids, and columns).

| Sample | Geometrical Porosity (%) | Porosity of the Foam Cells (%) | Fmax (N) | Compressive Strength (MPa) |
|---|---|---|---|---|
| Freeze Foam | 80 ± 0.5 | 76.1 ± 1.4 | 101 ± 53 | 0.9 ± 0.5 |
| Hybrid Foam | 74.4 ± 0.5 | 69.9 ± 0.9 | 2641 ± 452 | 23 ± 4 |
| VPP Column | 16.5 ± 0.7 * | | 3199 ± 831 | 31 ± 8 |
| Curasan | 55 ± 2 * | | 693 ± 89 | 3 ± 0.4 |

* Archimedes method.

Freeze Foams and hybrid foams exhibited similar porosity. However, the hybrid's compressive strength was 25 times higher (23 MPa) than the Freeze Foam alone (0.9 MPa). Surprisingly, the VPP columns alone showed an even higher compressive strength. Those values lie, however, within the standard deviation. It must be noted that the standard deviation was quite large. There were microdefects leading to failures in the macrostructure and/or the loaded surface was not plane, leading to varying forces upon contact with the compression stamp. The Curasan component provided the lowest porosity of all CaP scaffolds and showed much lower compressive strength than the hybrid foam (roughly one-seventh).

For interpretation of the simulation results, the maximum principal stress was considered because of the known brittleness of the support structure's ceramic material. Tensile load cases are critical for ceramics. Results of the first load case (Figure 10a) showed the largest maximum principal stress at the ringed segments. Maximum tensile stress appeared at the bottom surface of the top ring at around 61 MPa. Its origin can be assumed by the expansion of the rings by given external loads. This leads to an increase of a tangential

component of normal stress in the ring. For the first load case, a reaction force of 780 N was considered. In experiments, a structural failure occurred at an average load of 3200 N for the support structure. This corresponds to 800 N for one-quarter of the structure. Therefore, reaction force was increased for the second load case, up to 806 N (results displayed in Figure 10b). The largest maximum principal stress reached 80 MPa. Maximum tensile stress appeared at the same location, similar to the first load case. However, in the middle region of the columns (front), a tensile stress suddenly evolved, which was likely due to the buckling sensitivity of the support structure. At a certain uniaxial load, the structure will collapse because of buckling if the tensile strength of the bulk material is larger compared to this. In the third load case, the reaction force was increased to 809 N. Results are shown in Figure 10c. In this load case, the maximum tensile stress appeared in the middle of the columns at 128 MPa. The maximum principal stress was also high in the ring segments but not at this level. This led to the conclusion that, in third load case, the structure started to collapse by buckling. In Figure 11, a comparison of the three load cases for the maximum principal stress is shown. This figure clearly demonstrates the buckling sensitivity of the support structure. Between the first and the second load cases, the axial force was increased by 26 N.

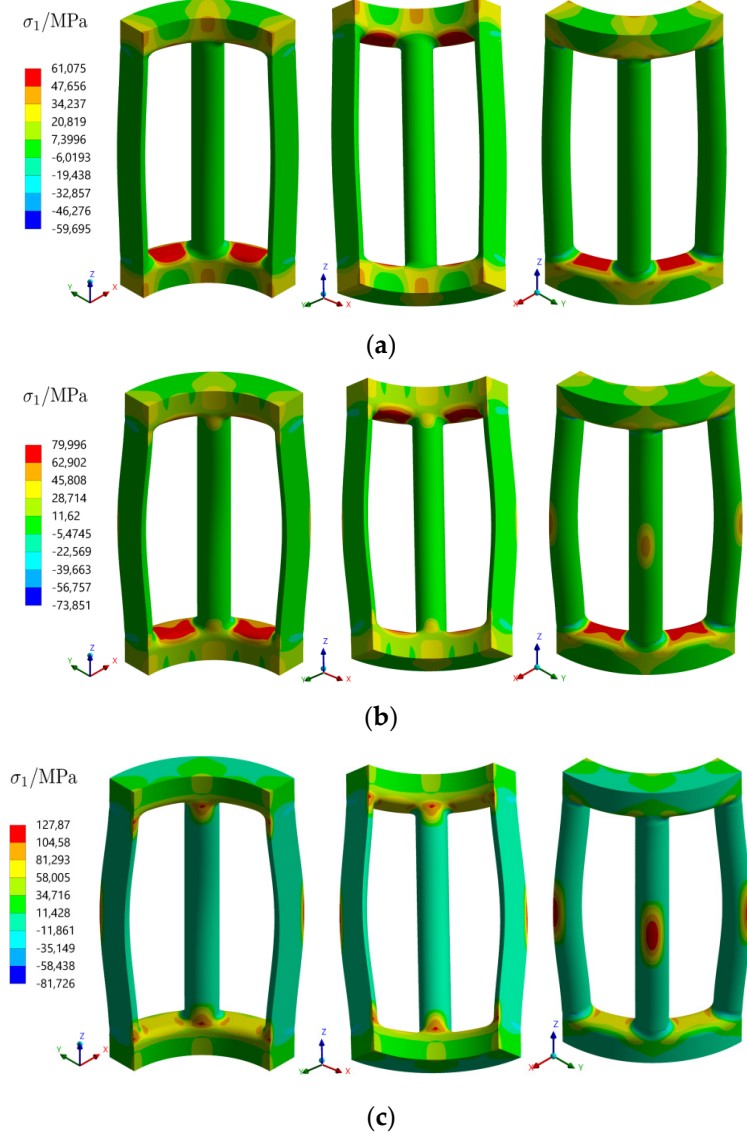

**Figure 10.** Results of analysis for the maximum principal stress: (**a**) @780 N load; (**b**) @806 N load; (**c**) @809 N load.

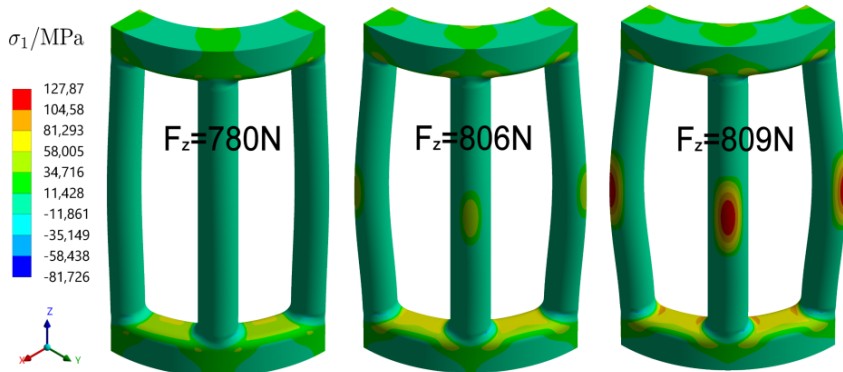

**Figure 11.** Comparison of uniaxial load sets (deformation shown in true scale).

This contributed to an expected increase of the maximum principal stress in the ringed segments but also led to an increase of tensile stress in the column of the structure. From the second to the third load case, a further small increase of axial force of 3 N significantly changed the load conditions. In the center region of the columns, maximum principal stress grew proportionally. The location of the maximum tensile stress changed from the ringed segments to the center region of the columns.

In Figure 12, experimental observed defects of the broken columns are shown. The red marked defects (primary defects) show those defects that initially occurred during experimental compression testing when the structure collapsed; the blue marked defects show the secondary defects that followed.

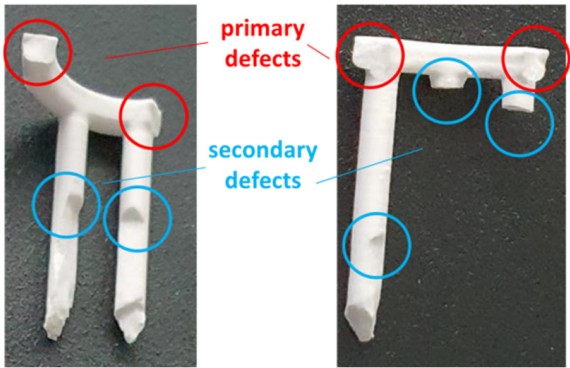

**Figure 12.** Experimental observed defects at the column structure.

The experimental failure pattern shown above leads to the assumption that the support structure collapsed by exceeding the tensile strength of the VPP-manufactured material, which corresponds to the FE analysis (first load case), and not by buckling, because cracks appeared at locations of maximum tensile stress in the ringed segments. However, some aspects must be considered. First, the simulation showed that the critical buckling loads appeared in the middle of the columns (third load case), which is very close to the experimental testing conditions (i.e., secondary defects shown in Figure 12). Therefore, complete structural failure by buckling should not be excluded. Second, the distribution of micropores in the VPP-manufactured column structure was considered homogenous for the simulated model. However, manufactured structures probably also include microdefects leading to the observed failure pattern.

Third, technological tolerances (by 3D printing, sintering, etc.) can lead to geometric imperfections (e.g., flatness of the ringed cross-section), which induce critical tensile stresses. Such effects were not included in the described FE model. At this stage of the presented research, there is still opportunity for adjustment if indeed microdefects appear, e.g.,

adjusting CerAM VPP exposure parameter, suspension parameters, debinding/sintering regime or the VPP design. By neglecting these aspects, a tensile strength of our manufactured material with 5% porosity could be obtained by further optimizing the VPP process. The tensile strength of a *β*-TCP 5% porous material should lie between 61 MPa and 80 MPa and take the mentioned restrictions into account.

### 3.3. In Vitro Biocompatibility

3.3.1. Live/Dead Assay

Human osteoblastic MG-63 cells were counted using Image-J (Fiji, Version 1.52 h), through which the cell number/mm$^2$ of living and dead cells was determined. Figure 13 shows representative samples with live/dead staining of the inner surface of the 500 µm scaffold as compared to the Curasan control after 3, 7 and 10 days. Long-term studies (>4 weeks) were not assessed.

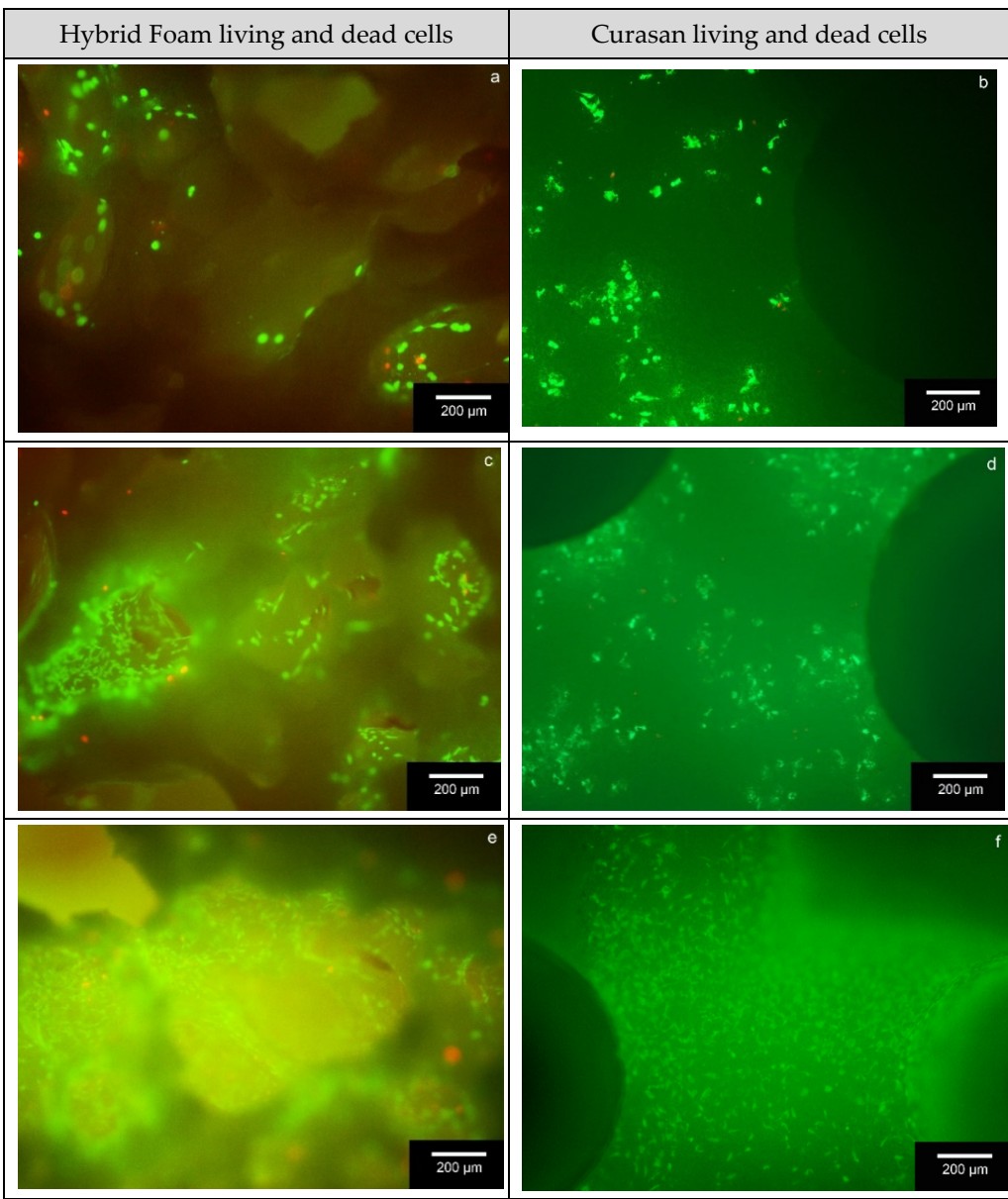

**Figure 13.** *Cont*.

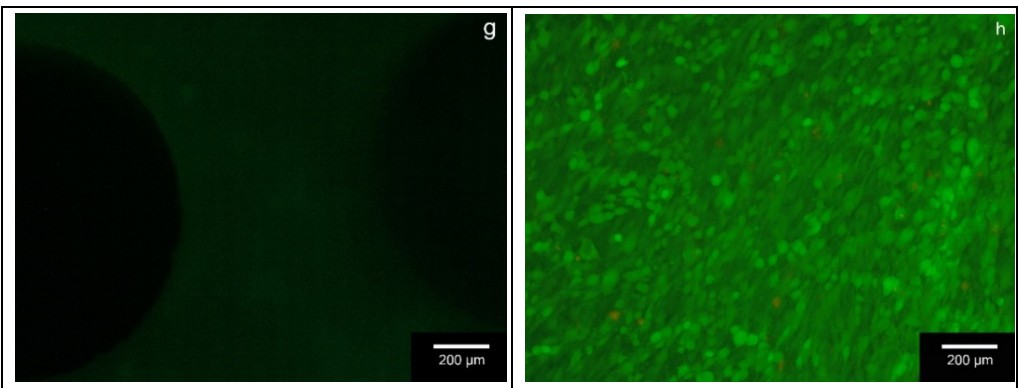

**Figure 13.** Living/dead cells on the inner surface of the ceramic; 500 μm scaffold after three days ((**a**): Hybrid foam; (**b**): Curasan), seven days ((**c**): Hybrid foam; (**d**): Curasan) and 10 days ((**e**): Hybrid foam; (**f**): Curasan); (**g**): Auto-fluorescence of the ceramics; (**h**): Thermanox membrane (pos. control, 10 days); white bar = 200 μm. Green indicates living cells; red indicates dead cells.

Quantitative results of the number of living and dead cells per mm$^2$ is shown in Figure 14a,b, respectively. The number of living cells increased over the course of the experiment in both the scaffold and the Curasan control, with no significant differences between our scaffold and the Curasan control.

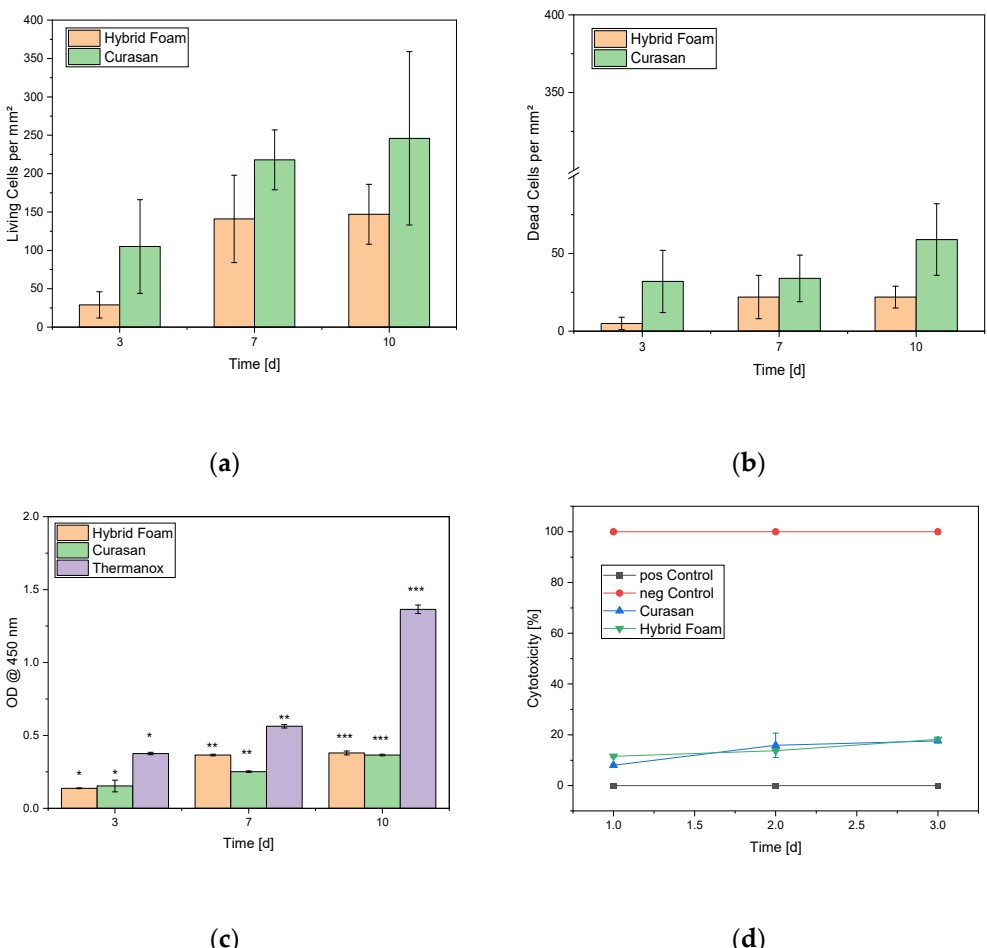

**Figure 14.** Overview of the biocompatibility tests: cell counts of (**a**) living cell numbers and (**b**) dead cell numbers per mm$^2$ on the materials after 3, 7 and 10 days; (**c**) WST assay to demonstrate proliferation of MG 63 on the samples. Means and controls were statistically compared to assess the material effect. Significances set at $p < 0.05$ are assigned the same symbol. (**d**) Cytotoxicity of hybrid foams compared to the Curasan control; pos. Control = cells, neg. Control = TritonX.

### 3.3.2. Cell Proliferation Assay

Figure 14c shows that the growth rate of the cells on the scaffold and Curasan control compared to the growth of cells on a Thermanox cover slip as a positive control. The growth rate of the cells on the scaffolds in the cell culture plates increased only up to seven days and stagnated thereafter, while the cells on the Thermanox cover slip continuously proliferated. No significant difference in cell proliferation was observed between the hybrid foam and the Curasan control.

### 3.3.3. LDH Assay

The cytotoxicity for both the hybrid foam and the Curasan control was slightly above the positive control (cells on the Thermanox cover slip) and very clearly below the negative control (Triton X), with no significant differences between our scaffold, the Curasan control and the positive control noted. The graphs of cytotoxicity over time were nearly congruent for the Curasan control and the Hybrid Foam (see Figure 14d).

### 3.3.4. GIEMSA Staining

In the GIEMSA staining, it was evident that the MG-63 cells only colonized the inner sponge area of the hybrid foams after 3 and 10 days, but only sporadically on the surface of the CerAM VPP shell (see Figure 15). Once again we saw mainly complete material and a form fit but also a gap between the ring structures and the Freeze Foam (Figure 15, upper right-hand side.)

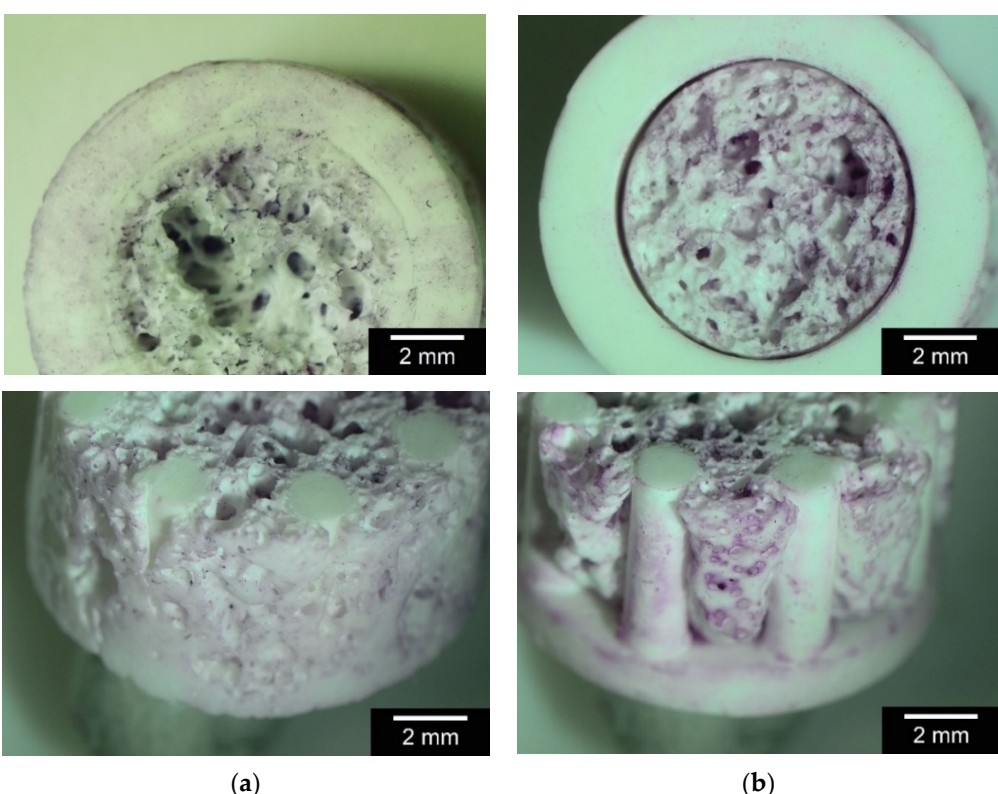

(**a**)  (**b**)

**Figure 15.** GIEMSA staining after (**a**) 3 and (**b**) 10 days; top and side view.

### 3.4. In Vivo Studies

Figure 16 displays manufactured TCP Freeze Foams (porosities between 83–85%, with an average of 84%), TCP hybrid scaffold halves (porosities around 80%) and single $ZrO_2$ Freeze Foams (porosities between 70–72%, with an average of 71%).

a.  Clinical Examination

After implantation, all rats had a score of "0" overall. This score means, clinically, we had no signs that the surgery or the implants had negative influences on the rats. However, the evaluation of histological samples (fine needle aspiration and explanted implantation area) had not been commenced. However, for all of the presented in vivo results, the authors think that the gathered serum parameters are more important regarding our following in vivo assessments than histological analyses would be at this stage of the research. We base our argument on studies of Trevisani et al. [33]. The main findings of this study were that the agreement between chronic histological kidney damage (CKD) and CKD staging was poor. In fact, about 30–40% of patients with CKD stage 3 had mild or no lesions in the histological evaluation (Chronicity Score = 0–1), whereas 7 to 10% of cases with CKD stage 1 (eGFR > 90 mL/min/1.73 m$^2$) had moderate or even severe histological lesions (Chronicity Score $\geq$ 3). Moreover, different patients with the same eGFR values may have had either severe (Chronicity Score $\geq$ 3) or no histological damage (Chronicity Score = 0) (eGFR = estimated glomerular filtration rate).

b.  Serum Parameters

The serum parameters of day 0, 2, 7, and 14 were measured by the accredited laboratory of the Clinic for Ungulates of the Veterinary Faculty of the University of Leipzig. These results were evaluated in a box-plot diagram to adjust them according to the physiological parameters as described in Charles River 2008 [34] and Boehm et al. [35] (Figures 21 and 22). Regarding these analyses, a photometric measurement (extinction determination) was executed. The photometric method is applicable for multi-species analyses [36]. Since no references were sent by the clinic, we had to compare the determined values, especially the creatinine and urea values, with the literature references of Charles River [34] and Boehm et al. [35].

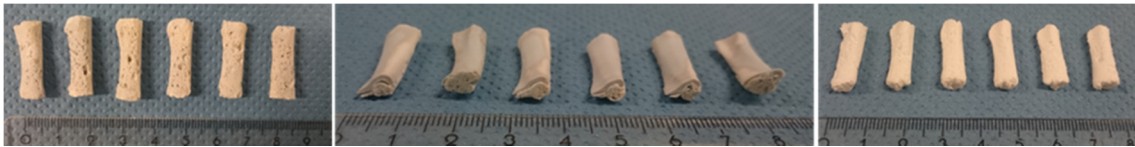

**Figure 16.** Comparison of the different manufactured scaffolds: left—TCP freeze foams, middle—TCP hybrid scaffold halves and right—ZrO$_2$ Freeze Foams.

### 3.4.1. ALT

It is clearly visible that all alanine transaminase (ALT) values of all rats on the surgery and the control days are greater than the references in the literature. Furthermore, there is one higher aspartate aminotransferase (AST) value in the negative control group, while all the other AST values of the other four rats and the median of this group stay in the reference area over the experimental time. Although all ALT and AST values in one rat of the negative control were this high over the entire time period, there is no sign of liver damage as proven by Gamma-Glutamyl Transferase (GGT), as the liver-specific value is located well within the reference in all rats all over the experiment. However, the chosen physiological ALT reference may not be specific enough for rats, or it might be increased by food containing a higher amount of proteins. Furthermore, the indicated references in different literature sources varied between 25 U/L and 163 U/L [35] in the mean for ALT and from 26 U/L to 155 U/L in mean for AST. They also varied based on age and sex. That is why we took a critical view of these parameters according to their values. In addition, it is well known in veterinary medicine that ALT and AST are also produced in other organs, e.g., in muscles and kidneys both are produced quickly in response to medical/toxic agents, and values three times the upper limit attract attention in practice. As the rats had such a medical supply during anesthesia, analgesic treatment, and anesthesia during surgery, and analgesics were injected, the increase of ALT and AST might be a result

of this treatment (see Figure 17). However, it is noteworthy that, with the exception of ALT levels on day 14, AST and ALT concentrations were within the physiological range in all of the experimentally treated groups.

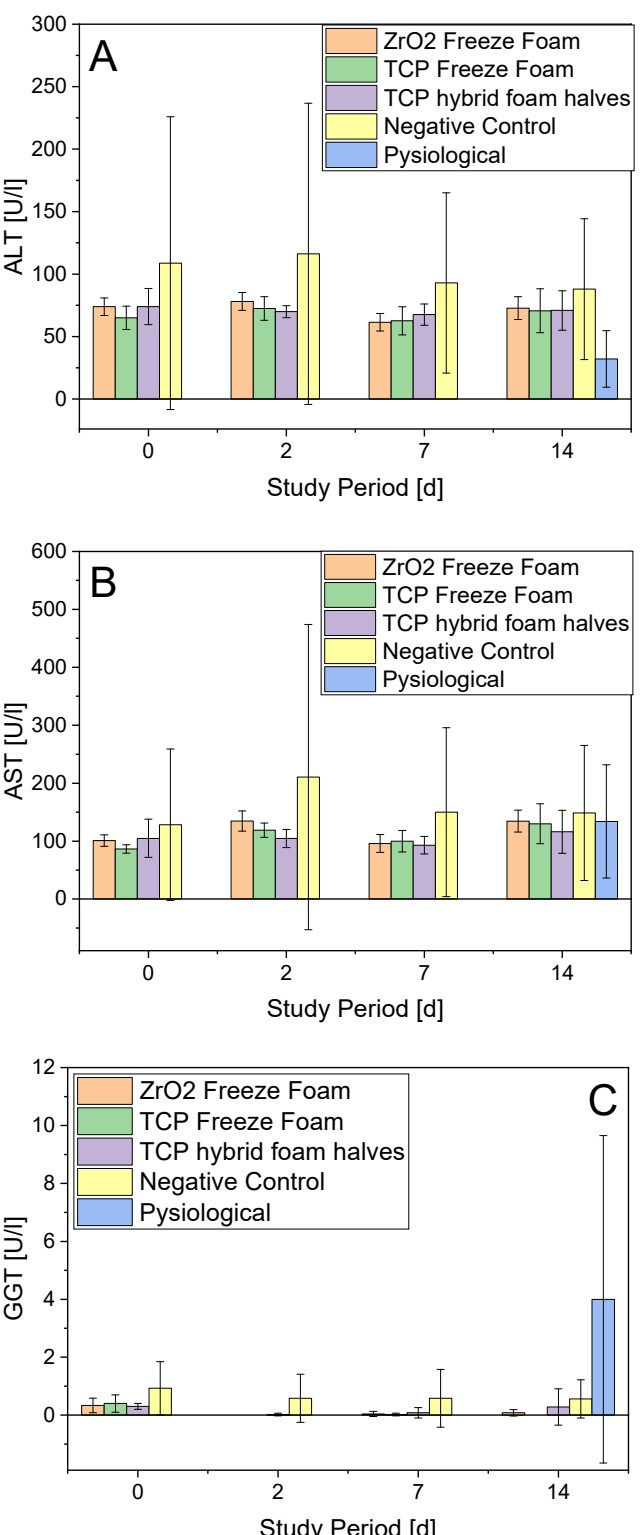

**Figure 17.** Course of (**A**) ALT, (**B**) AST and (**C**) GGT over the study period of 14 d; for clarity, the physiological control values were only entered for day 14. Physiological values taken from Giknis et al. [34] (ALT, AST) and Boehm et al. [35] (GGT).

### 3.4.2. Creatinine

Kidney creatinine (crea, long-term) and urea (short-term) values stayed under the references in all rats of all groups at all time points (Figure 18). As values depend heavily on the method used for their determination, the engaged laboratories have their own references. Since no references were sent by the clinic as stated above, we compared the determined values of creatinine and urea with references [34,35]. This might be an explanation for the lower values. However, as the kidney values did not show any increases over the time, this shows that the implants did not have a negative influence on the kidneys. Nevertheless, regarding proof of a non-toxic effect, long-term additional histological examinations of the removed kidneys, in line with the 3R principles, must be executed.

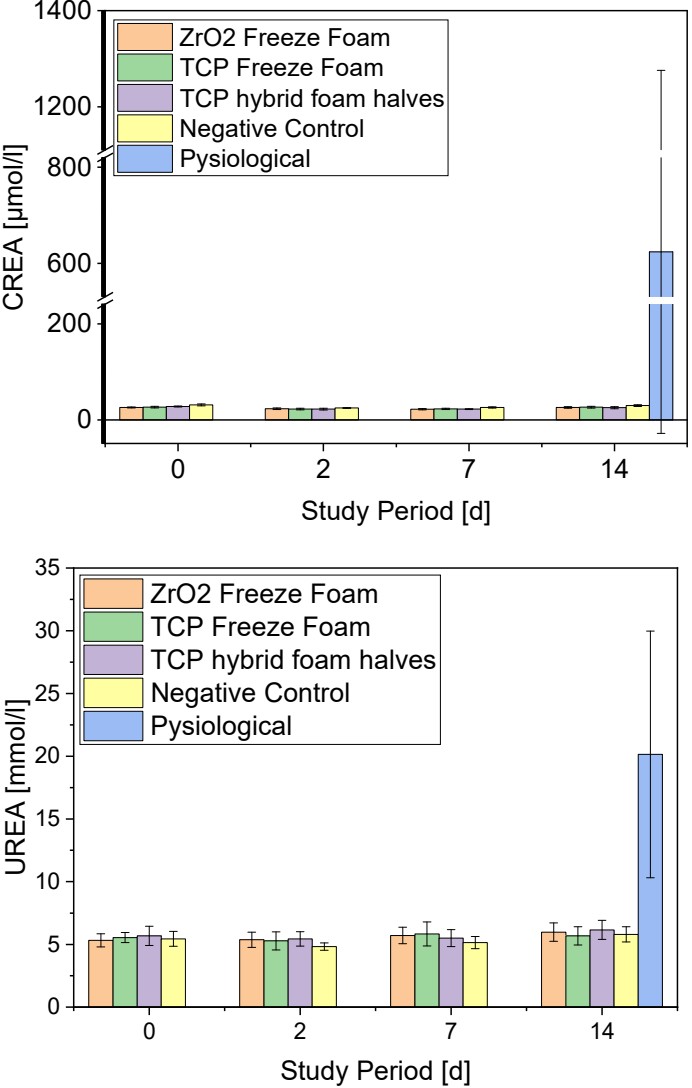

**Figure 18.** Concentrations of creatinine and urea from days 0 to 14. For clarity, the physiological control values were only entered for CREA and UREA at day 14. The physiological values for creatinine (CREA) and UREA were taken from Boehm et al. [35].

### 3.4.3. Necropsy

All rats in all treatment groups showed no macroscopic alterations of the spleen, kidney or liver during the necropsy. One rat treated with the $ZrO_2$ foams showed a minimally increased spleen, which was probably caused by post-mortal blood congestion. All rats in all treatment groups showed a slight to moderate increase of the regional lymph nodes, which is likely due to the resorption and healing processes after surgery.

### 3.4.4. Implant Parameters

As written above we judged the implants according to different semi-objective parameters. One of them was their ability to be vascularized or rather the adherence and ingrowth of tissue to the implants after 14 days. Vascularization is important for the supply of oxygen, nutrients, the transport of metabolic products and immune cells. As the goal is to help bone to grow into this pattern, a high vascularization results in a high metabolic rate and growth factors in the area. As shown in the following figure, the more porous the artificial bones the more tissue grows into them (84% TCP Freeze Foam > 81% TCP hybrid foam halves > 71% ZrO$_2$ Freeze Foams). The zirconium oxide group showed less tissue adherence than the TCP specimen (Figure 19A). Additionally, we noted a non-adherence of host tissue to the part of the hybrid foam halves that were made by CerAM VPP for the artificial *corticalis*. As we do not only want vessels to grow into the implants but ultimately, host tissue to replace the implant and later giving stability, we evaluated the removability of tissue from the implants after 14 days. It is obvious that not only the porosity seems to be an important factor for the surrounding tissue but also the material. As expected, TCP was more integrative than zirconium oxide (Figure 19B). We also observed that parts of broken implants were held together by the immigrated tissue giving them additional stability. On the other hand, this adherent growing can be seen critical in case of a removal of an implant for instance due to incompatibility or failure. Though this growth was very invasive, we detected no signs of macroscopic fibrosis, capsular formation, inflammation, or calcification in the implantation area. An additional statement could be given after the evaluation of histological samples (fine needle aspiration and explanted implantation area). As the biodegradable implants are developed to replace bone in the short to midterm, they have to provide enough stability until the hosts own bone material is calcified. Therefore, we also looked at the scaffolds loss of stability and the tendency to break after 14 days (Figure 19C). In necropsy we did not find evidence of a broken implant in the ZrO$_2$ group, likely due to a slightly decreased porosity as well as its general material properties), one broken implant in the TCP Freeze Foams and one nearly broken implant in the hybrid bone, whereby the fracture was located at the connection between porous and additively manufactured shell part. These tendencies likely reflect the material and porosity properties.

### 3.4.5. In Vivo Conclusion

From the macroscopic and clinical point of view, and according to incompatibility and toxicity, we had no sign that any of the implants, independent of the material, the manufacturing or the handling before implantation, negatively influenced the results of this oriented and leveled study. This well-founded statement is based on proven literature references as discussed above and as demonstrated in [36,37]. In accordance, our in vivo results are unobtrusive. In that regard, we can make a recommendation that the scaffolds be developed further as a result of their vascularization/tissue ingrowth tendency, which is an important factor for an implant in the muscoskeletal system. The TCP Freeze Foams are the most promising scaffolds for a use in artificial trabecular bones according to the determined parameters in the study. The TCP hybrid foam halves (artificial *corticalis* case) showed insufficient connection of additively manufactured parts to the tissue. In accordance with the in vitro analyses, where the cells only sporadically colonized/attached to the CerAM VPP-manufactured columns, the tissue did not adhere to the CerAM VPP shell structure but only to the porous artificial *spongiosa* acting Freeze Foams (see Figure 15). Roughness measurements indicated that there were clear differences between the manufactured components, with the additively manufactured one likely being too smooth for cell attachment (see Figure 9). However, they still may be good candidates for further development, considering the fact that the shell part is very stable, likely for a long time, and thus this implant could potentially allow bridging of very big/long bone defects. However, a solution must be found to enhance cell attachment capability (e.g., chemical and/or physical surface modification and/or adding porosity). Alternatively, the support structure case might be chosen.

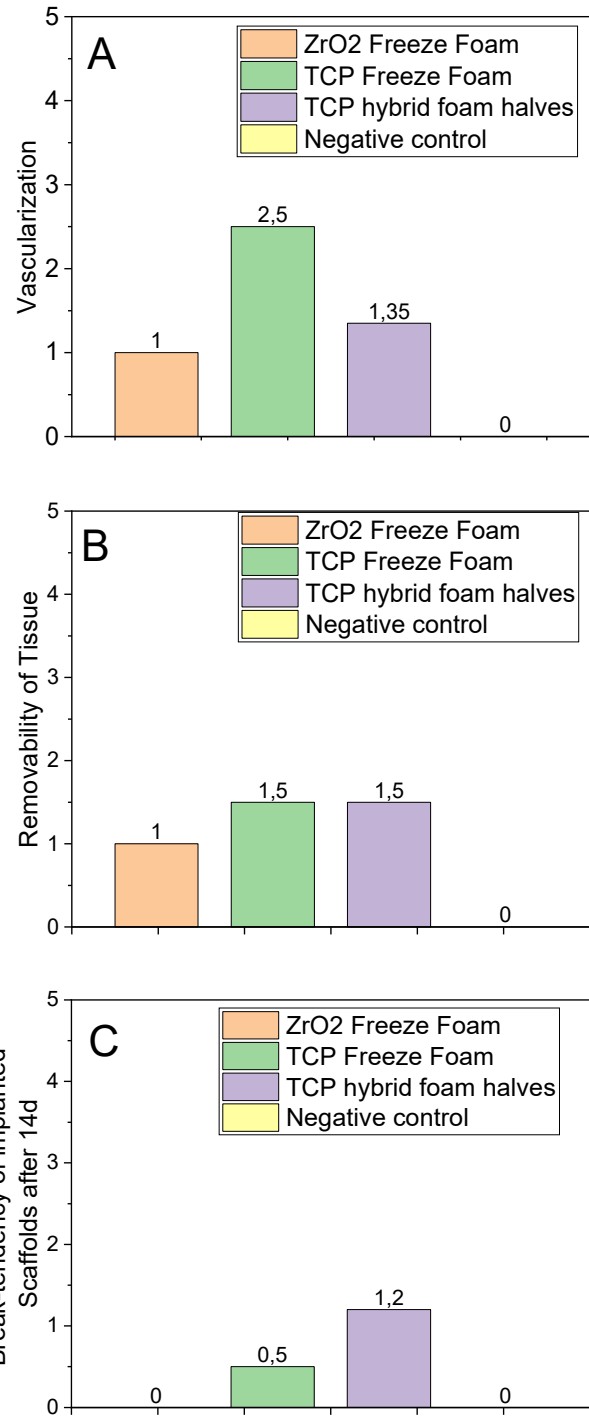

**Figure 19.** Implant parameters after 14 days: (**A**) tissue adherence/ingrowth into the artificial bones; (**B**) removability of tissue from the implants; (**C**) break-tendency of implanted scaffolds.

## 4. Discussion

An ideal engineered tissue scaffold for the regeneration of load-bearing bones should possess appropriate mechanical functions to provide structural support, share the biomechanical load, and distribute stress that stimulates bone growth and remodeling. Due to excellent biocompatibility, calcium phosphate scaffolds have been successfully used for non-load-bearing bone restoration in recent decades. Current bioceramic scaffolds cannot re-establish massive load-bearing bones. The mechanical properties of materials decrease with increasing porosity and pore size. For optimal new bone building, scaffolds normally



require an interconnected macroporous structure, with a high porosity of over 90% and a pore size ranging from 100 to 1000 μm. Such porous constructs typically have low mechanical properties. Therefore, in this study, porous bone-like foams were fitted around/in a customized additively manufactured support structure to manufacture bioceramic hybrid foams that are mechanically much more stable than single porous components. The scaffolds were made of β-TCP and were analyzed in terms of their biocompatibility and mechanical behavior. As a result, the authors postulate that these complex structural hybrids, due to the combination of load-bearing support and porous cell-ingrowth allowing interior growth, will eventually allow the manufacturing of bone-mimicking, mechanically stable implants.

### 4.1. Microstructural and Mechanical Characterization

All manufactured Freeze Foams and hybrid foams showed the microstructural characteristics necessary for use as potential bone replacement materials and implants, including macro and micro/mesoporosity of the right size as well as interconnectivity. Manufactured hybrid foams showed connected/joined porous and dense sections similar to a real bone. However, more parts must be analyzed in order to obtain a general overview of the success rate of materials as well as form fit. On the other hand, it is yet not clear to what extent apparent gaps between additively manufactured and porous components have influenced the mechanical as well as biocompatibility behaviors. More research needs to be conducted. Regarding the already enhanced compressive strength, for possible further improvements and increased failure tolerance, the column structure should be reengineered. For example, this can be done by adding a further ring in the structure's middle or making the columns meet each other in the center of the structure (reduction buckling length). Optimization of the VPP process would further result in less porosity and fewer microdefects, assuming that the same material and thermal treatment is applied. However, through improving loads, the biodegradability will most likely decline. A balance between good biocompatibility and sufficient mechanical strength must be found. The maximum failure load was 29 ± 9.0 N for the commercially available Curasan β-TCP ceramic [38], 693 ± 85 N for the Curasan cube, which served as a control, and 2641 ± 452 N for the hybrid foam. The hybrid's failure load was 91 times higher than the commercially available Curasan ceramic and four times higher than the control. There were comparable values for compressive strength: 23 ± 4 MPa for the hybrid foam and 24 ± 6 MPa for the Curasan β-TCP ceramic. Freeze Foams and hybrid foams exhibited similar porosity. However, the hybrids compressive strength was 25 times higher (23 MPa) than the Freeze Foam alone (0.9 MPa). The additively manufactured bioceramic support structures made the porous spongious structures mechanically more stable. Bone is structurally complex and hierarchically designed. Cortical bone is stronger and stiffer in comparison to trabecular bone. The material behavior of cortical bone is anisotropic. The compressive moduli of cortical bone along the longitudinal direction (193 MPa) are greater than those along the transverse direction (133 MPa) [39]. The compressive moduli of trabecular bone is 50 MPa. Trabecular bone is a highly porous material with anisotropic mechanical properties. Due to its high porosity versus that of cortical bone, the mechanical properties of trabecular bone are determined primarily by its porosity. The mechanical properties of the bone are thus still higher than the measured hybrid foams, which still only have a strength support structure and no surrounding *corticalis*. With the help of FE analysis, we were able to approximate mechanical loads appearing in the structure and to analyze and predict failure mechanisms that then also occurred in the mechanical tests.

### 4.2. Biocompatibility

Regarding the biocompatibility experiments, the most noticeable aspect was that the cells did not attach to the VPP-manufactured parts. There was a clear gap present. We now need to work out whether that gap was correlated with a possible mismatch/non-material fit between the VPP part and the Freeze Foam, as shown before. Alternatively, the cells

might behave like this because the VPP part is too smooth to "hold" onto. Despite this, the cells clearly attached to the foam surface and even grew into the Freeze Foam, as proven in in vitro and in vivo results. In the WST-1 experiment, the ceramics and hybrid foams and the control group showed comparable vitality values and a constant cell growth over the examined period of time. These results are congruent with the ones seen in the live-dead assay. The hybrid foam sample showed a similar high biocompatibility comparable to the Curasan ceramic. This is not surprising, since both samples consist of β-TCP. We were able demonstrate the high biocompatibility of β-TCP in various studies in the past [15,16,40]. However, the proliferation values for the Curasan sample were slightly lower than those of the hybrid foam, which may be due to the fact that cells generally prefer a structured surface [41,42]. In addition, the Curasan sample had a lower porosity. In terms of cytotoxicity, both samples, the hybrid foam and the Curasan sample, are on an equal footing, with partly even congruent curves. This was also not surprising, as both studies involved β-TCP. In line with our previous studies, β-TCP is non-cytotoxic [16,40,43].

Regarding the manufactured zirconia Freeze Foams, we showed that the specific porous structure/pore morphology resulting from the Freeze Foaming process allows tissue ingrowth independent of the bioceramic materials used.

## 5. Conclusions

As mentioned in the introduction and in the discussion, previous studies show that β-TCP is a performing bone replacement material, not only as pure material and shaped by conventional methods (e.g., freeze drying) but also as composite material (e.g., with polymers) and shaped by additive manufacturing. The resulting compressive strength is always relatively low, ruling out load-bearing clinical indications/bone defects. In contrast, our results show that we can address load-bearing bone defects using the same material as previously reported, by advancing known composites to become complexly shaped structural composites that not only unite the structural features of a real bone (dense and porous sections) but also reach similar and improved compressive strengths (of trabecular bone [44,45]), while at the same time providing degradability as given by the material. By fine-tuning the support structure design and working on composite materials to develop new structural and material composites for potential bone replacements, we might be able to further develop mechanical properties, aligning our approach with a variety of bone defects, especially long-bone and load-bearing ones. Offering the same biocompatibility, the bioceramic hybrid foams have significant mechanical advantages over the Curasan benchmark. The hybrid's failure load is 91 times higher in comparison to the commercially available β-TCP ceramic. To summarize, the compressive strength of the bone-mimicking hybrid bones was significantly enhanced, while high biocompatibility was maintained as proven on the Curasan material. At present, the BMBF-funded project "Hybrid-Bone" (03VP07633) is in progress, which builds on these results and strives for the evaluation and validation of materials, processes and hybrid scaffolds for use as compressive-strength-enhanced biodegradable jaw-bone replacements. Within the framework of this project, the comprehensive bone-forming performance tests of similar scaffolds, with a focus on hybrid foams, are carried out in animal models. The authors hope to report on these results soon.

**Author Contributions:** Conceptualization, M.A.; Data curation, M.A., S.H.L., C.S., C.F., D.W., E.S.-F. and M.S.; Investigation, C.F., D.W. and E.S.-F.; Methodology, M.A. and H.O.M.; Project administration, M.A.; Software, C.F.; Supervision, M.A.; Validation, S.H.L., C.S., C.F., D.W. and E.S.-F.; Visualization, S.H.L. and M.S.; Writing—Original draft, M.A.; Writing—Review and editing, M.A. and M.S. All authors have read and agreed to the published version of the manuscript.

**Funding:** This research received no external funding.

**Institutional Review Board Statement:** The animal study (file number 47/18) was reviewed and approved by the Landesdirektion Leipzig, Germany. The approval date was January 15th in 2019 and the study was conducted according to Directive 2010/63/EU.

**Informed Consent Statement:** Not applicable.

**Acknowledgments:** This contribution originally was intended to be published together with Anke Bernstein, for Musculoskeletal Biomaterials of the Research Center for Tissue Replacement, Regeneration and Neogenesis (G.E.R.N.) and the Deputy Director of the G.E.R.N in the Department of Orthopaedics and Trauma Surgery at the University Hospital Freiburg in Breisgau, Germany. With our deepest regrets, Anke died in a mountaineering accident at the end of June 2021. Bernstein's research revolved around biomaterials-based therapies for repair and regeneration of musculoskeletal tissues. Bernstein was a dedicated member of the German Society for Biomaterials and held the role of the President since 2019. Anke, you will be remembered as a friendly person with a sharp mind, a great sense of humor and lots of laughter. Thank you, indeed, for walking some of the way together. We offer Anke Bernstein's family, friends and colleagues our sincere condolences on her passing. Many thanks as well to Melanie Lynn Hart for spell-checking this contribution.

**Conflicts of Interest:** The authors declare no conflict of interest.

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
