# Peer review of "Mechanically Stable β-TCP Structural Hybrid Scaffolds for Potential Bone Replacement"

_jcs, doi:10.3390/jcs5100281_

Round 1

Reviewer 1 Report

This is such a nice job, The authors were so careful and done decent justifying experiments.  Pleased to accept in its present form

The only concern,  i have, 24 Figures of this manuscript, Try to combine the appropriate images together and reduce the numbers. 

Author Response

Many thanks dear Reviewer No 1. for you very kind assessment of our work. This is very much appreciated. We heeded your well-thought suggestion and reduced the number of figures to 19.

We very much hope that this new version will suffice.

Best regards

The authors of this work

Reviewer 2 Report

The manuscript by Ahlhelm et al. describes the manufacturing of a composite B-TCP scaffold made by CerAM VPP and freeze foaming methods. Morphological, mechanical, in vitro and in vivo biological characterizations are reported. 
The statement at page 22, lines 675-677 is the synthesis of the whole work: "...it is yet not clear to what extent apparent gaps between additively manufactured and porous components have an influence on the mechanical as well as biocompatibility behavior". The Authors should clarify several issues if they want to publish something new, relevant and useful for the whole scientific community. More specific suggestions are reported below.

Figure 4-6 show results, not experimental data. Thus it is suggested to move them in the Results section.

Page 7, lines 266-272: The geometry, morphology and overall shape of the commercial samples used as comparison control do not match those of the developed samples. This mismatch surely has an impact on in vitro biocompatibility studies. The Authors should comment on that point.

Page 14, lines 480-482: the Authors state that their samples may have micro defects. How will the Authors face this issue? Are there approaches to overcome the manufacturing drawbacks? something should be discussed herein, or at least it is suggested to add a sentence to refer to the discussion section (i.e. page 22, lines 678-684).

Page 18, lines 540-543, lines 568-569: why do the Authors describe histological samples assessment if they still have no results? It is strongly suggested to remove this sentence, as well as the relevant experimental part (i.e. page 9, lines 369-372). 

Section 3.4.1 describes the assessment of ALT, AST and GG values, but the same Authors state that the collected results might be affected by analgesic treatment (this Reviewer agrees). Why the Authors report these data if they are not relevant to evaluate the composites compatibility? Even the high standard deviations should be condidered to discard the mentioned results.

Page 19, lines 577-580: a statement like this could not be reported in a rigorous scientific paper. If histological examination is required to discuss the obtained results, it is essential that the Authors integrate the manuscript with histology. The same applies to page 21, lines 626-628 and to page 23, lines 725-727.

Author Response

Dear Reviewer No. 2, many thanks for this comprehensive review. Subsequently, we answer all your questions and add the input to the paper accordingly. Starting with your first suggestion: the structure was changed accordingly.

Further point-to-point response is reported in the attached document.

We very much hope that this new version will suffice.

Best regards

The authors of this work

Reviewer 3 Report

Reviewer’s report

Title: Mechanically Stable β-TCP Structural Hybrid Scaffolds for potential Bone replacement

Comments and Suggestions for Authors

 In general, paper is interesting for reading, I found some editing errors (line: 555, 611), no dot or dot in the wrong place. The current manuscript can be accepted after considering the following comments:

The introduction to the article lacks a strong reference to previously obtained research results presented in other articles. It is important because it will help to understand the added value of conducted research. In the section describing the scaffolding made of TCP, please refer to other works carried out earlier, e.g.  “3D printed β-TCP bone tissue engineering scaffolds: Effects of chemistry on in vivo biological properties in a rabbit tibia model”; doi: https://doi.org/10.1557/ jmr.2018.233 or https://doi.org/10.1002/jbm.a.35711. I think that the authors should clearly highlight that the possibility of obtaining functional hybrid structure (porous and solid) is an important novelty presented in the article.

Section 2_2. Please explain why the model with 8 supports, columns was chosen? I have some doubts here whether such a small number of supports will reflect the properties of the actual outer layer of the bone? Was the model consisting of 8 supports introduced intentionally, if so, why? The FEM analysis shows a sharp increase in the value of bending stresses on the basis of the analyzed model. It would probably be possible to reduce the stress value by increasing the number of supports in the developed model.

Equation 2: Please explain the abbreviations used in the equation. They are self-explanatory, but should be included in the article.

Unfortunately, in Figures 18, 22 and 23, measurement deviations and axis descriptions are difficult to read. 

I think that after minor corrections your work is ready to publish.

Best regard

Reviewer

Author Response

Many thanks dear reviewer No 3. We appreciate to review and included your valuable suggestions (please see document). 

We very much hope that this new version will suffice.

Best regards

The authors of this work

Round 2

Reviewer 2 Report

Regretfully, in this reviewer opinion, the Authors efforts to improve the quality of their work are not enough for publication in Journal of composites science.

Author Response

Dear Reviewer No 2,

We feel sorry that our point-to-point answers and scientifically proven arguments did not meet your expectations. We feel that we answered you correctly and we are supported in our opinion and statements by the view of two other reviewers. Hopefully, we will meet your expectations the next time you are reviewing our work.

Please find our full rebuttal (original of Review No 3) in the attached document.

Best regards

The authors of this work

Reviewer 3 Report

Dear Authors,

thank you for correcting the manuscript. In my opinion, the manuscript is ready for publication as it stands. 

Best regards

Reviewer

Author Response

Dear Reviewer No 3,

We are glad that our improved work met your high standards and expectations.

Best regards and many thanks

The authors of this work

This manuscript is a resubmission of an earlier submission. The following is a list of the peer review reports and author responses from that submission.

Round 1

Reviewer 1 Report

In this study, a natural bone-like structure was fabricated by freezing foaming technology, and the mechanical characteristics of different structures were analyzed through finite element. At the same time, the actual mechanical test of the scaffold also verified similar results. However, this study still has several shortcomings:
1. The article only discusses the cytocompatibility of the scaffold and the toxicity in vivo, and does not conduct relevant research on the bone formation performance of the scaffold. I hope the author can explain the relevant bone formation performance;
2. The picture in Figure 16 is too small and can be enlarged appropriately;
3. Please explain why you choose MG63 cells, why not choose other cell lines or primary cells;
4. The picture in Figure 21 is not clearly labeled;

Author Response

Dear reviewer 1,

many thanks for reading our contribution and for providing your valuable time in helping us publishing our newest scientific achievements. In the following we state your comments and directly answer them in the hope of finding your agreement:

  1. The article only discusses the cytocompatibility of the scaffold and the toxicity in vivo, and does not conduct relevant research on the bone formation performance of the scaffold. I hope the author can explain the relevant bone formation performance;
  • Indeed, we are working at the bone formation performance of similar scaffolds right now. This is the project Hybrid-Bone. Because I was expecting the significance, I was already hinting at this project in the conclusion. I added some sentences that we are planning/doing bone forming performance tests right now in the mentioned project (Page 21, line 729. I just didn’t want to make too much of advertisement.
  1. The picture in Figure 16 is too small and can be enlarged appropriately;
  • When online, the images should be zoomable. Regarding the review, we thought that the editor/journal included the images in high res (what they are) in the supplement for you to look at. However, we optimized and merged them for a better viewing experience in the revised manuscript.  

  1. Please explain why you choose MG63 cells, why not choose other cell lines or primary cells;
  • Other cells e.g., human osteoblasts, are not necessarily better, because they, for example, tend to differentiate. The advantage of the commonly used MG63 cells is that they always behave the same (e.g., no change of morphology) and are therefore, reliably comparable.
  1. The picture in Figure 21 is not clearly labeled.
  • Thank you for the advice. We took care of it in adapting the figure caption.

Reviewer 2 Report

The authors report biocompatibility and bioactivity of bone constructs made from combining additive manufacturing with conventional freeze foaming process.

  1. The study is incomplete, without any histological analysis of the in vivo studies. I suggest completing them before making speculative conclusions.
  2. Section 2 states that hydroxyapatite has been used as a raw material. I do not understand why that has been called β-TCP through the manuscript.
  3. I don’t agree that sintering changes hydroxyapatite to β-TCP. The authors need to augment this claim with XRD and FTIR data for the used scaffolds or powder.
  4. The choice of U-test for the statistical comparison is not correct. I suggest redoing the statistical significance using ANOVA.
  5. Compressive strength data suggests they are not strong enough to be used for load-bearing implants. I suggest calling these implants low/no load-bearing scaffolds.
  6. Figure 16 is of pretty poor quality. I suggest processing the images and removing the background from the images to allow clarity. I also suggest the authors merge the live and dead cells together in one image to show any overlap (if any) in the live and dead cells.
  7. Figure 16 shows that the cells on the hybrid foam are rounded without any observable cell spreading, unlike in the positive control. Together with stagnation in cell viability after 10 days suggests toxicity in a long-term culture. I suggest the authors perform long-term cell culture studies or provide ion degradation data (by ICP-MS measurements) of the scaffolds over a 4-week time-period.
  8. Figure 17: I suggest the cell viability should be plotted as a percentage of live cells vs dead cells. The way it is demonstrated makes it difficult for the reader to conclude anything.
  9. I am not sure that the in vivo section makes much sense without histological analysis.

Author Response

Dear reviewer 2,

many thanks for reading our contribution and for providing your valuable time in helping us publishing our newest scientific achievements. In the following we state your comments and directly answer them in the hope of finding your agreement:

The study is incomplete, without any histological analysis of the in vivo studies. I suggest completing them before making speculative conclusions.

  • Indeed, we stated several times that no histological analysis were made yet. We even stated that this is why the results must be regarded critically (page 21, line 633). At this stage we won’t get the analyses within meaningful time. But with the explicit statement that the results must be viewed critically because no histological analyses are available yet there is no reason not show the in vivo results in the current status. Based on the given results the data shown is not speculative. It is only incomplete in deriving robust and extensive evidence for successful in vivo trials. By the way: what is successful? Our animals didn’t die (in the given period) because of our structures. In addition, we only implanted into rat’s flank not into bone. In vivo trials can be manifold. Everything is mentioned and the reader can make his opinion based on our (not hidden) data. Are you willing to accept the already stated critical notion of the in vivo studies? If not, can you suggest wordings to your liking?

Section 2 states that hydroxyapatite has been used as a raw material. I do not understand why that has been called β-TCP through the manuscript.

  • On page 7, line 245 we state: „It must be noted that after sintering the initial hydroxyapatite has changed to β-TCP.“ Since compressive strength, in vitro and in vivo tests were made with sintered structures it is the TCP that shows its properties and was analyzed. 

I don’t agree that sintering changes hydroxyapatite to β-TCP. The authors need to augment this claim with XRD and FTIR data for the used scaffolds or powder.

  • To my knowledge quite usually does initial HA powder and shaped parts of it transfer to TCP during heat treatment (which is a common step for ceramic material). That is known and can be referred to in:
  1. Ravaglioli und A. Krajewski, Bioceramics, p. 57, 1992.
  2. Locardi, V. Pazzaglia, C. Gabbi und B. Profilo, Biomaterials, Nr. 44, p. 437, 1993.
  3. Legeroz und J. Legeros, An Introduction to Bioceramics, p. 139, 1993.
  4. Gottschling, R. Kohl, A. Engel und H. Oel, Bioceramics: Materials and Applications, p. 201, 1994.
  5. Wang, „Investigation on High-temperature decomposition characteristic of hydroxyapatite,“ in International Conference on Nano/Molecular Medicine and Engineering (NANOMED), pp. 65-70, 18.-21. October 2009.
  6. Chal und B. Ben-Nissan, „Hydroxyapatite - Thermal Stability of Synthetic Hydroxyapatite,“ International Ceramic Monographs, Bd. 1, pp. 79-85, 1994.

C.-J. Liao, F.-H. Lin, K.-S. Chen und J.-S. Sun, „Thermal decomposition and reconstitution of hydroxyapatite in air atmosphere,“ Biomaterials, Nr. 20, pp. 1807-1813, 1999.

  • I am not sure what other readers make of it when I am listing those references now, stating that HAp usually degrades to TCP. It is usually known. If you insist, I will add them though. ALTERNATIVELY, I will do the following: I made those XRD measurements on similar specimen back in 2012 with the same initial material and similar Freeze Foams. Therefore, I briefly stated: Amongst many other references reporting about the transition of HA to TCP during heat treatment, similar Freeze Foams with the same initial HA powder were analyzed via XRD in previous works [22] showing the HAp to TCP transformation.

The choice of U-test for the statistical comparison is not correct. I suggest redoing the statistical significance using ANOVA.

  • Thank you, we now included ANOVA.

Compressive strength data suggests they are not strong enough to be used for load-bearing implants. I suggest calling these implants low/no load-bearing scaffolds.

  • In this case I would be glad to have references clearly and explicitly clarifying what is load bearing and what is not. For example, jawbone trabecular bone withstands compressive strengths around 3,9 MPa. We reach 23 MPa. It must be load-bearing then.

Figure 16 is of pretty poor quality. I suggest processing the images and removing the background from the images to allow clarity. I also suggest the authors merge the live and dead cells together in one image to show any overlap (if any) in the live and dead cells.

  • Images are in fact high resolution. We thought the editor/the journal would provide you with the full high-res pics in the supplement for you to look at. We now merged them and improved the viewing experience. In the end though, we are limited by the editor’s/journal’s restriction.

Figure 16 shows that the cells on the hybrid foam are rounded without any observable cell spreading, unlike in the positive control. Together with stagnation in cell viability after 10 days suggests toxicity in a long-term culture. I suggest the authors perform long-term cell culture studies or provide ion degradation data (by ICP-MS measurements) of the scaffolds over a 4-week time-period.

  • Indeed, we would very much like to do such measurements. Unfortunately, we don’t have the equipment available. In addition, probably 70% of papers published don’t show long-term studies (what is long term?). Are they unrightfully published? As stated above: Everything is mentioned and the reader can make his opinion based on our (not hidden) data. However, I added the sentence: Shown data and results do not represent long-term studies (> 4 weeks).

Figure 17: I suggest the cell viability should be plotted as a percentage of live cells vs dead cells. The way it is demonstrated makes it difficult for the reader to conclude anything.

  • We are of the opinion that exactly the opposite is the case. The plot of the cell viability as percentage is not effective because the total number of cells, as in a representation of cells/mm², is not included. With the same distribution of living/dead cells, it would make no difference whether there are many or few cells because the percentage is the same. However, it should be shown that the number of cells increases over time. So we combined both Origin Graphs into a new one.

I am not sure that the in vivo section makes much sense without histological analysis.

  • In our opinion it does. Based on scientific measurements it is shown that the novel material and (hybrid) structures were not rejected by the animals. Especially the facts 1. we had tissue ingrowth and 2. that no encapsulation took place hint at good in vivo properties (are these successful in vivo results?)

Round 2

Reviewer 1 Report

The evaluation of MG63 cells is not a good description of the osteogenesis performance of the scaffold, and the article does not have more in vivo and in vitro data to support the author's point of view. For the evaluation of the osteogenic ability of the scaffold, I suggest that it is necessary to provide the relevant evaluation of osteoblasts and the evaluation of the in-situ osteogenic ability in vivo, especially the ability to repair long bone defects.

Reviewer 2 Report

The authors have not responded to many of rhe queries raised or performed the experiments suggested to them in the last round of review.